# From Experience to Strategy: Empowering LLM Agents with Trainable Graph Memory

## Abstract

Large Language Models (LLMs) based agents have demonstrated remarkable potential in autonomous task-solving across complex, open-ended environments. A promising approach for improving the reasoning capabilities of LLM agents is to better utilize prior experiences in guiding current decisions. However, LLMs acquire experience either through implicit memory via training, which suffers from catastrophic forgetting and limited interpretability, or explicit memory via prompting, which lacks adaptability. In this paper, we introduce a novel **agent-centric, trainable, multi-layered graph memory** framework and evaluate how context memory enhances the ability of LLMs to utilize parametric information. The graph abstracts raw agent trajectories into structured decision paths in a state machine and further distills them into high-level, human-interpretable strategic meta-cognition. In order to make memory adaptable, we propose a reinforcement-based weight optimization procedure that estimates the empirical utility of each meta-cognition based on reward feedback from downstream tasks. These optimized strategies are then dynamically integrated into the LLM agent's training loop through meta-cognitive prompting. Empirically, the learnable graph memory delivers robust generalization, improves LLM agents' strategic reasoning performance, and provides consistent benefits during Reinforcement Learning (RL) training.

## 1 Introduction

LLM-based agents are rapidly advancing the frontier of automated task execution, particularly in open-ended environments that demand long-horizon reasoning, strategic tool use, and adaptation from experience (Yao et al., 2022; Gao et al., 2023; Chai et al., 2025). While these agents demonstrate strong capabilities in decomposing and tackling complex tasks, their decision-making processes remain unstable, often resulting in inefficient action sequences, repeated mistakes, or even complete task failure (Singh et al., 2023). A central challenge lies in empowering agents not only to act, but to continuously learn and adapt by extracting insights from past successes and failures.

Methods for enabling LLMs to better leverage prior experience can be broadly categorized into two paradigms. The first type is **implicit memory**, typically formed through training procedures such as RL, meaning that LLMs encode syntactic structures and semantic relations into parameter space (Li et al., 2025b; Bai et al., 2022). A more flexible alternative is **explicit memory** leveraged via contextual prompting, which improves performance by injecting context directly into the input without modifying model weights. (Xu et al., 2025; Chhikara et al., 2025; Zhao et al., 2024).

However, both paradigms suffer from fundamental yet contrasting limitations. While explicit memory facilitates transparency by making reasoning steps externally visible through prompts, it often lacks adaptability and struggles to generalize beyond specific tasks or contexts. Conversely, implicit memory enables generalization via training, yet its black-box nature renders the contribution of specific past experiences untraceable. Furthermore, encoding knowledge directly into parameters often incurs information loss and is vulnerable to catastrophic forgetting. This dilemma motivates our central research question: *Can dynamic and structured explicit memory be used to provide a stable, learnable prior that meaningfully improves implicit policy learning in LLM agents?*

This paper introduces a novel **agent-centric, trainable, multi-layered graph memory** framework and investigates its utility within RL paradigms. First, we map raw agent trajectories into canonical

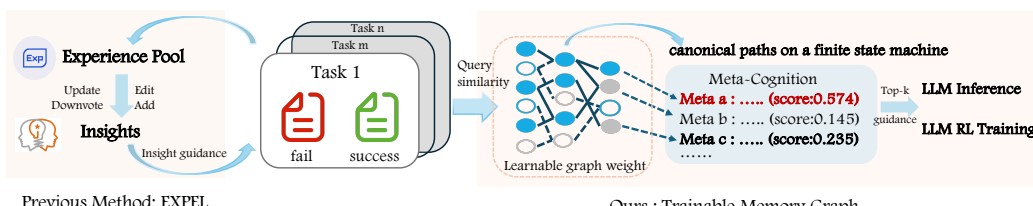

Figure 1: Our method and existing approach Expel (Zhao et al., 2024).

paths on a Finite State Machine(FSM), which allows us to extract high-level, reusable *meta-cognitive strategies*. Second, we build a trainable graph memory whose edge weights can be optimized via reinforcement learning, enabling the system to learn which strategies are truly useful for downstream tasks. Finally, the dynamic graph serves as an *explicit policy prior*, selectively injecting high-quality strategies into the agent's context during training. Empirical results across seven diverse question-answering benchmarks demonstrate that our framework delivers strong gains in both cross-task generalization and final task performance.

Our main contributions are threefold:

- We propose a novel **agent-centric graph memory framework** that abstracts low-level agent trajectories into canonical FSM paths, enabling the distillation of **high-level, generalizable meta-cognitive** strategies.

- We design a **reinforcement-driven weight optimization mechanism** that dynamically calibrates the utility of graph weights, ensuring the graph prioritizes strategies with high empirical utility.

- We demonstrate that integrating this graph memory as an **explicit policy prior** significantly enhances RL efficiency and final task performance.

Ultimately, this work presents a unified framework for adaptive agents that continuously learn and reason from their own evolving experiences.

## 2 RELATED WORK

### 2.1 LLM AGENTS AND PLANNING WITH EXTERNAL TOOLS

LLM agents increasingly incorporate external tools to overcome reasoning limitations and expand their problem-solving capabilities. Early prompt-based approaches, including ReAct (Yao et al., 2022) and WebGPT (Nakano et al., 2021), demonstrate how agents can interleave reasoning and acting, embedding tool calls directly in the generation trace. Building on these foundations, Search-o1 introduces agentic RAG that dynamically retrieves knowledge during reasoning. Building on these foundations, Search-o1 (Li et al., 2025a) advances tool-augmented reasoning by enabling agents to autonomously decide when to invoke search tools during multi-step problem solving. Recent research has proposed more sophisticated coordination mechanisms using RL-based training (Sun et al., 2025; Zheng et al., 2025; Song et al., 2025). Search-R1 (Jin et al., 2025) represents a breakthrough RL framework that trains LLMs for alternating reasoning and search, enabling autonomous query generation and real-time information retrieval during step-by-step reasoning. Other recent approaches include optimized reward designs (Wang et al., 2025; Qian et al., 2025) and strategic tool integration (Feng et al., 2025), with frameworks like RL-Factory (Chai et al., 2025) accelerating research in this domain.Despite these advances, the lack of explicit long-term memory for reusable tool-use patterns leaves deciding when and which tools to invoke as a key bottleneck. To address this limitation, we propose a differentiable graph-based memory system that encodes past decision paths into reusable strategic priors, enabling agents to systematically learn and generalize planning strategies across domains.

## 2.2 MEMORY ARCHITECTURES AND STRATEGIC LEARNING

Recent research has increasingly explored how to extract strategic knowledge and meta-cognition from agent experience. Reflexion (Zhang et al., 2023) equips agents with self-verbalized feedback to refine future behavior, while Expel (Zhao et al., 2024) identifies reusable reasoning trajectories to guide subsequent decisions. MEM1 (Zhou et al., 2025) and MemAgent (Yu et al., 2025) adapt memory usage over long-horizon tasks. A-MEM (Xu et al., 2025) builds dynamic memory notes that evolve with new inputs, Zep (Rasmussen et al., 2025)and HopRAG (Liu et al., 2025) construct logic-aware graphs to facilitate retrieval.

However, these methods typically apply graph structure in a static manner and lack mechanisms to assess or refine the utility of memory components. G-Memory (Zhang et al., 2025) demonstrates how hierarchical graph-based memory can evolve by assimilating new collaborative trajectories, enabling systems to leverage cross-trial knowledge and learn from prior experiences progressively. Pan & Zhao (2025)focus on whether different forms of memory can enhance reasoning. Xiong et al. (2025) investigate long-term memory evolution.While prior memory methods often rely on static storage or task-specific designs, they lack mechanisms for evaluating and refining strategies. In contrast, we propose a trainable graph-based memory that supports utility-aware strategy selection and reinforcement learning–driven updates, enabling generalizable and adaptive decision-making.

## 3 PRELIMINARIES

### 3.1 HETEROGENEOUS GRAPH STRUCTURE

Graphs provide a natural formalism for modeling structured dependencies among diverse entities. A heterogeneous graph (Zhang et al., 2019)can be defined as

$$\mathcal{G} = (V, E, \mathcal{O}_V, \mathcal{R}_E, C),$$

where $V$ denotes the set of nodes, $E \subseteq V \times V$ denotes the set of directed edges, $\mathcal{O}_V$ denotes the set of node types, $\mathcal{R}_E$ denotes the set of relation types, and $C$ is the collection of node contents. Each edge $e = (u, v, r) \in E$ specifies a relation of type $r$ from node $u$ to node $v$.

Connectivity in $\mathcal{G}$ is represented by node-type adjacency matrices

$$A^{xy} \in \{0, 1\}^{|V_x| \times |V_y|}, \quad (x, y) \in \mathcal{O}_V \times \mathcal{O}_V,$$

where $\mathcal{V}_x$ and $\mathcal{V}_y$ denote the sets of nodes of type $x$ and $y$, respectively. An entry $(A^{xy})_{ij} = 1$ indicates that node $i$ of type $x$ is connected to node $j$ of type $y$. This formulation emphasizes the structural dependencies across different node types.

To enable learning, each $A^{xy}$ is coupled with a weight matrix $W^{xy}$, so that propagation is governed by the weighted operator $A^{xy} \odot W^{xy}$. Thus, structure defines feasible paths, while weights determine effective information flow. Formally,

$$\mathbf{H}_y = \sigma\left((A^{xy} \odot W^{xy})^\top \mathbf{H}_x\right),$$

where $\mathbf{H}_x$ are source values and $\sigma(\cdot)$ denotes an activation function.

### 3.2 LLM AGENTS WITH TOOL-AUGMENTED REASONING

The interaction between a LLM and external tools can be formalized as a structured multi-turn decision process (Chai et al., 2025). At each time step $t$, the agent observes

$$s_t = (q, h_{1:t-1}), \quad a_t \sim \pi_\theta(a_t \mid s_t).$$

where $q$ is the user query and $h_{1:t-1}$ the dialogue or reasoning history, then generates an action $a$ which may correspond to internal reasoning, a tool invocation, or answer generation, using a protocol with tags such as `<think>`, `<tool_call>`, and `<answer>`.

The process continues until either the tag `<answer>` is generated or the agent has issued up to a maximum of $K$ tool invocations. A trajectory $\tau = (s_1, a_1, o_1, \ldots, s_T, a_T, o_T)$ yields reward $R(\tau)$

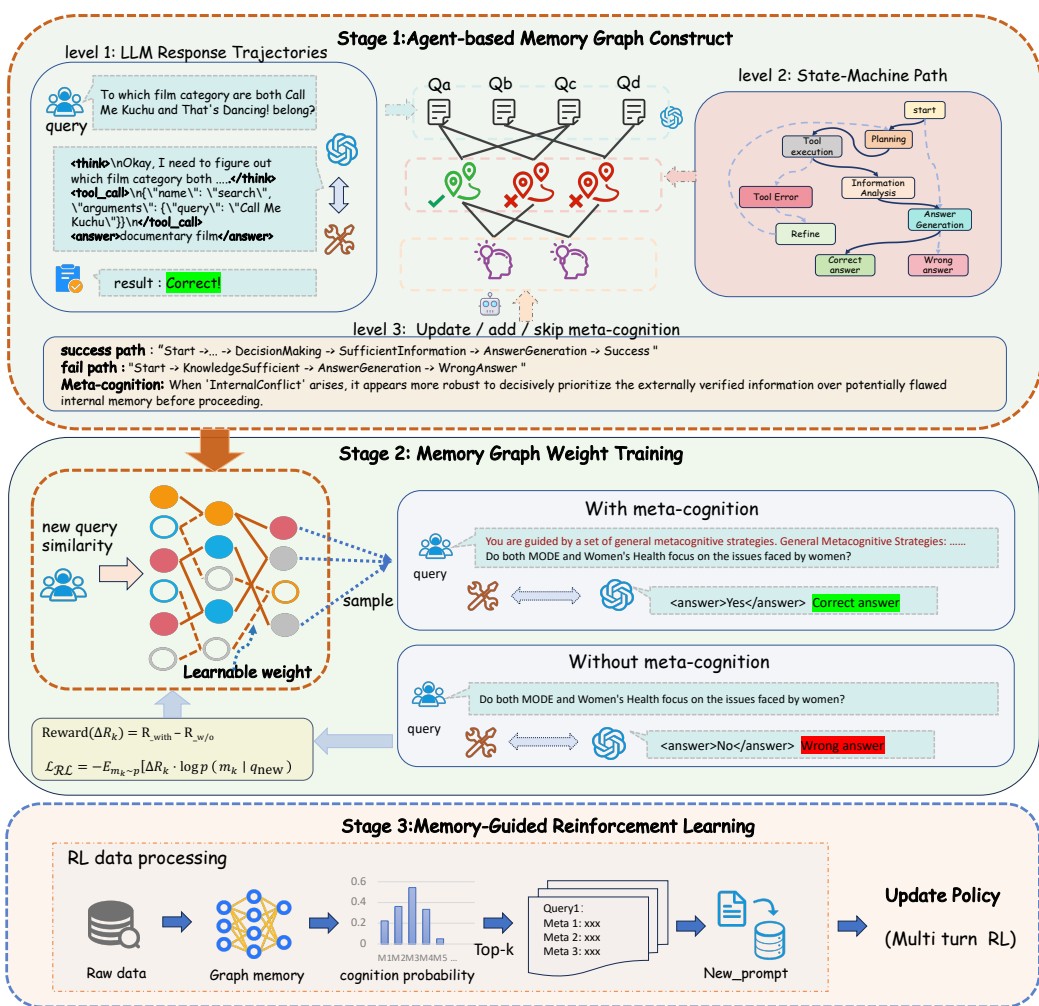

Figure 2: The framework of the trainable memory graph. Stage 1 builds a graph from LLM trajectories, encoding queries, decision paths, and meta-cognition. Stage 2 estimates strategy utility via counterfactual rewards and updates graph weights. Stage 3 injects top-k strategies into RL training for policy optimization.

, where $o_t$ denotes the observation, i.e., the environment's response (e.g., tool outputs) if $a_t$ is a tool call, and the policy is optimized via

$$J(\theta) = \mathbb{E}_{\tau \sim \pi_\theta}[R(\tau)], \quad \nabla_\theta J(\theta) \approx \mathbb{E}_\tau \left[ \sum_{t=1}^{T} \nabla_\theta \log \pi_\theta(a_t \mid s_t) \, \hat{A}_t \right].$$

## 4 METHOD

In this section, we detail our proposed method in three stages. First, we describe how to construct a memory graph that encodes decision trajectories and strategic principles. Second, we present the learning framework for optimizing the weights within this memory graph. Finally, we explain how this structured memory is integrated into the RL training process to guide agent behavior and improve learning efficiency. The overall process of our method is shown in the Figure 2.

### 4.1 STAGE 1: HIERARCHICAL MEMORY GRAPH CONSTRUCTION

**Memory Graph Structure.** We instantiate a heterogeneous graph $\mathcal{G} = (V, E)$ with a three-tier reasoning hierarchy consisting of three node types: queries, transition paths, and meta-cognitions (see Figure 2 Stage 1):

- **The Query Layer** $\mathcal{Q}$: Each query node $q_i$ stores the input, sampled trajectories, and outcome labels. Because a single query may yield multiple rollouts, $q_i$ connects to several transition paths via edges $(q_i \to t_j)$.

- **The Transition Path Layer** $\mathcal{T}$: Each path node $t_j$ represents a canonical cognitive trajectory produced by mapping raw reasoning traces into FSM paths. This abstraction removes surface-level linguistic noise and exposes the underlying decision structure shared across tasks.

- **The Meta-Cognition Layer** $\mathcal{M}$: Each meta-cognition node $m_k$ captures a reusable strategic principle distilled from one or more transition paths. These principles reflect why reasoning patterns succeed or fail (e.g., conflict resolution) and serve as domain-agnostic behavioral heuristics.

The complete construction process is illustrated in Figure 3. Formally, the graph connectivity is encoded by the bipartite adjacency matrices augmented with learnable weights $w_{qt}$ and $w_{tm}$ respectively. This yields a directed acyclic graph enabling information to flow from concrete experiences toward increasingly abstract strategic knowledge.

$$A^{q \to t} \in \{0,1\}^{|\mathcal{Q}| \times |\mathcal{T}|}, \quad A^{t \to m} \in \{0,1\}^{|\mathcal{T}| \times |\mathcal{M}|},$$

**Finite State Machine.** Given that raw reasoning trajectories are often laden with stylistic noise and redundancy, we define an FSM $\mathcal{S} = (S, A, T)$ to map each trajectory onto a standardized sequence of cognitive states, such as `StrategyPlanning` or `InternalConflict`. This abstraction effectively distills noisy natural-language reasoning into canonical decision paths $t_j$. Crucially, this transformation filters out execution-level details to expose the underlying decision structure, enabling robust comparison across tasks. Refer to Appendix C for the full FSM specification.

**Meta-Cognition Induction and Update.** For each query, we sample trajectories $\{\tau_1^{(i)}, \ldots, \tau_N^{(i)}\}$. If both successful and failed trajectories exist, their FSM paths are contrasted to extract a high-confidence causal meta-cognition explaining the divergence. When only failures occur, we retrieve semantically similar queries (using $\cos(\mathbf{e}_{q_i}, \mathbf{e}_{q_j})$) and borrow successful patterns from their transition paths:

$$\mathcal{M}^{\text{spec}}(q_i) = \bigcup_{q_j \in \text{TopK}(q_i)} \{m_k : t_j \in \text{SuccessPaths}(q_j),\ m_k \in \mathcal{M}(t_j)\}.$$

As new trajectories accumulate, the graph evolves. If the candidate insight aligns with an existing meta-cognition, the system refines that principle by integrating new evidence and sharpening its conditions of applicability. If the candidate reflects a previously unseen failure mode or reasoning strategy, the system creates a new meta-cognition node. Through this mechanism, the memory graph expands only when necessary while compressing redundant patterns into stronger, more generalizable principles. Concrete cases and prompt templates are shown in Appendix G.3.

The hierarchical structure thus abstracts low-level trajectories into reusable strategies. At inference, $\mathcal{G}$ functions as a structured policy prior guiding decision-making, while during training it provides supervision signals for reward-driven consolidation of meta-cognitive knowledge.

## 4.2 STAGE 2: TRAINABLE GRAPH WEIGHT OPTIMIZATION

The memory graph provides structural priors, but not all meta-cognition contribute equally. To adaptively capture their utility, we introduce a reinforcement-driven weight optimization procedure.

**Parameterizing the Graph for Utility Learning.** We parameterize $\mathcal{G}$ as a sparsely connected weighted network, where each edge is associated with a trainable coefficient reflecting its utility. Given query features $\mathbf{H}_{\mathcal{Q}}^{(0)}$, information propagates through the graph via weighted aggregation:

$$\mathbf{H}_{\mathcal{T}}^{(1)} = \sigma\left((A_{qt} \odot W_{qt})^\top \mathbf{H}_{\mathcal{Q}}^{(0)}\right), \quad \mathbf{H}_{\mathcal{M}}^{(2)} = \sigma\left((A_{tm} \odot W_{tm})^\top \mathbf{H}_{\mathcal{T}}^{(1)}\right),$$

which corresponds to the flow from the *query layer*, through the transition layer, and finally to the *meta-cognition layer* in Figure 2.

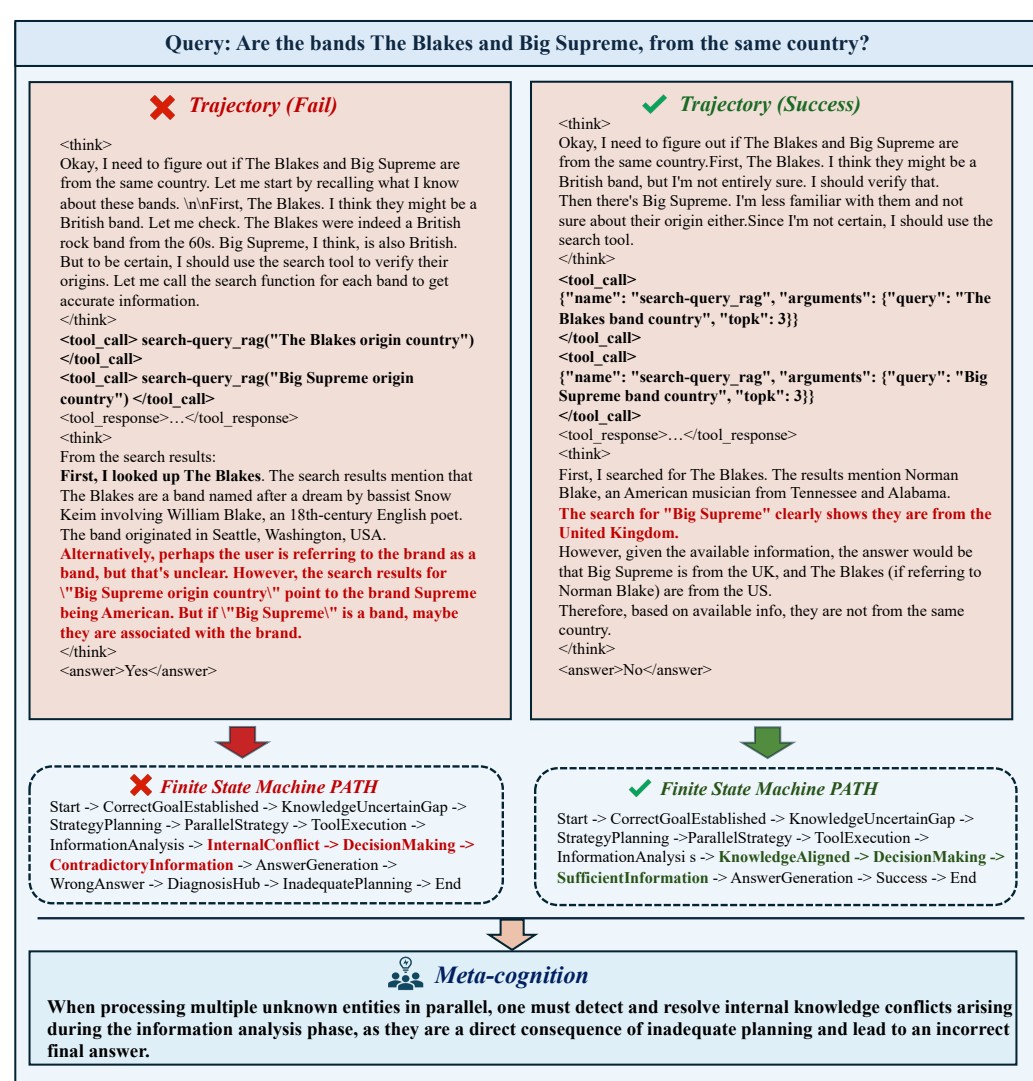

Figure 3: An example of graph construction.

In our formulation, a new query is represented by its similarity to historical queries in the graph, and the top-$k$ most relevant neighbors are selected to activate a task-specific subgraph $\mathcal{G}(q_{\text{new}}) = (\mathcal{Q}', \mathcal{T}', \mathcal{M}')$. Within this subgraph, a candidate meta-cognition $m_k \in \mathcal{M}'$ is sampled according to a relevance score $\rho(m_k)$, derived from the learned graph weights.

To estimate its empirical utility, we contrast two trajectories: one guided by $m_k$, which yields reward $R_{\text{with}}(m_k)$, and another without such guidance, yielding reward $R_{\text{w/o}}$. The resulting *reward gap* $\Delta R_k = R_{\text{with}}(m_k) - R_{\text{w/o}}$ is employed as a utility signal, quantifying the marginal contribution of $m_k$ to overall task performance.

**Policy Gradient-Based Weight Optimization.** The relevance score $\rho(m_k \mid q_{\text{new}})$ is computed by aggregating path strengths from historical queries and transitions leading to $m_k$:

$$\rho(m_k \mid q_{\text{new}}) = \sum_{q_i, t_j : q_i \to t_j \to m_k} \text{Sim}(q_{\text{new}}, q_i) \cdot w_{qt}^{(i,j)} \cdot w_{tm}^{(j,k)}.$$

Using a softmax over these scores, the selection probability $p(m_k \mid q_{\text{new}}) \propto \exp(\rho(m_k \mid q_{\text{new}}))$.

We apply the REINFORCE algorithm to optimize the weights:

$$\mathcal{L}_{\text{RL}} = -\mathbb{E}_{m_k \sim p}\left[\Delta R_k \cdot \log p(m_k \mid q_{\text{new}})\right].$$

A positive $\Delta R_k$ increases the relevance score and strengthens the supporting paths, while a negative $\Delta R_k$ decreases them, enabling the memory graph to refine itself over time.

### 4.3 STAGE 3: MEMORY-GUIDED POLICY OPTIMIZATION

Departing from prior works that leverage memory solely during inference, our framework explicitly integrates the structured memory into the *training loop*. Meta-cognitive strategies are dynamically retrieved from the optimized memory graph and incorporated into the agent's context, serving as high-level strategic priors that guide the RL process.

**Strategic Context Retrieval.** For each training instance $q_{\text{train}}$, we compute a relevance score for every meta-cognition node $m \in \mathcal{M}$. This score is derived from the aggregated weights of all paths connecting the corresponding query node to the meta-cognition node within the memory graph $\mathcal{G}$ (as formulated in Section 4.2). We then select the top-$k$ meta-cognitions $\{m_1, \ldots, m_k\}$ with the highest scores. This mechanism ensures that the guidance is not only relevant but also grounded in empirically successful past trajectories, as encoded by the learned edge weights.

The retrieved strategies are verbalized and prepended to the original query to form an augmented prompt, $\tilde{q}_{\text{train}} = \left[\, m_1, m_2, \ldots, m_k \,;\, q_{\text{train}} \,\right]$, this augmented prompt serves as the input to the policy network.

**Optimization Objective.** The agent's policy, $\pi_\theta$, is optimized to maximize the expected cumulative reward conditioned on the augmented context. We employ a policy gradient method, where the parameters $\theta$ are updated by minimizing the following loss function:

$$\mathcal{L}_{\text{RL+Mem}} = -\mathbb{E}_{a \sim \pi_\theta(\cdot \mid \tilde{q}_{\text{train}})}\left[R(a)\right].$$

This tight integration ensures that the policy does not learn in isolation but is continually guided by a dynamically evolving corpus of strategic knowledge. This allows the agent to effectively bootstrap its learning process from a distilled representation of past successes.

In practice, we adopt the Generalized Reinforcement Policy Optimization (GRPO) algorithm to optimize the memory-augmented policy. The GRPO loss can be written as:

$$\mathcal{L}_{\text{GRPO}} = -\mathbb{E}_t\left[\min\left(\frac{\pi_\theta(a_t \mid \tilde{q}_{\text{train}})}{\pi_{\theta_{\text{old}}}(a_t \mid \tilde{q}_{\text{train}})}\hat{A}_t,\; \text{clip}\left(\frac{\pi_\theta(a_t \mid \tilde{q}_{\text{train}})}{\pi_{\theta_{\text{old}}}(a_t \mid \tilde{q}_{\text{train}})}, 1 - \epsilon, 1 + \epsilon\right)\hat{A}_t\right)\right],$$

where $\hat{A}_t$ is the advantage estimator and $\epsilon$ the clipping parameter.

## 5 EXPERIMENT

### 5.1 DATASETS

To evaluate the effectiveness and generalizability of our approach, we conduct experiments on seven widely-used question answering datasets, covering both single-turn and multi-hop reasoning tasks. **(1) General QA Datasets**: We include Natural Questions (Kwiatkowski et al., 2019), TriviaQA (Joshi et al., 2017), and PopQA (Mallen et al., 2022), which consist of open-domain factoid questions requiring retrieval and basic reasoning capabilities. **(2) Multi-hop QA Datasets**: For more complex reasoning scenarios, we adopt HotpotQA (Yang et al., 2018), 2WikiMultiHopQA (Ho et al., 2020), Musique (Trivedi et al., 2022), and Bamboogle (Press et al., 2022), which require integrating information across multiple documents.

### 5.2 BASELINE EVALUATION

To comprehensively evaluate the effectiveness of our proposed method, we design experiments from two complementary perspectives: **(1) Direct Inference Impact:** We assess how the integration of

Table 1: Performance comparison across seven QA datasets in inference. [†]indicates in-domain datasets, while [⋆]denotes out-of-domain datasets. Percentages in Avg. column denote relative improvement over TIR.

| Methods | Avg. (↑ / ↓ vs. TIR) | General QA | | | Multi-Hop QA | | | |
|---|---|---|---|---|---|---|---|---|
| | | NQ[⋆] | TriviaQA[⋆] | PopQA[⋆] | HotpotQA[†] | 2wiki[⋆] | Musique[⋆] | Bamboogle[⋆] |
| **Qwen3-8B** | | | | | | | | |
| TIR | 0.334 (–) | 0.275 | 0.593 | 0.358 | 0.325 | 0.324 | 0.094 | 0.365 |
| Direct Inference | 0.269 (↓19.5%) | 0.200 | 0.519 | 0.191 | 0.230 | 0.275 | 0.058 | **0.410** |
| CoT | 0.252 (↓24.6%) | 0.209 | 0.512 | 0.182 | 0.223 | 0.271 | 0.055 | 0.308 |
| Raw Trajectory | 0.352 (↑5.4%) | **0.317** | 0.604 | 0.380 | 0.329 | **0.363** | 0.105 | 0.364 |
| A-MEM | 0.334 (0.0%) | 0.286 | 0.590 | 0.366 | 0.339 | 0.332 | 0.112 | 0.313 |
| EXPEL | 0.329 (↓1.5%) | 0.306 | 0.594 | 0.379 | 0.317 | 0.327 | 0.092 | 0.287 |
| Ours | **0.365(↑9.3%)** | 0.316 | **0.622** | **0.382** | **0.358** | 0.354 | **0.128** | 0.392 |
| **Qwen3-4B** | | | | | | | | |
| TIR | 0306 (–) | 0.298 | 0.581 | 0.351 | 0.268 | 0.281 | 0.077 | 0.290 |
| Direct Inference | 0.211 (↓31.0%) | 0.158 | 0.413 | 0.157 | 0.183 | 0.240 | 0.033 | 0.290 |
| CoT | 0.181 (↓40.8%) | 0.149 | 0.375 | 0.146 | 0.156 | 0.190 | 0.022 | 0.228 |
| Raw Trajectory | 0.325 (↑6.2%) | 0.310 | 0.558 | 0.379 | 0.282 | 0.344 | 0.076 | 0.327 |
| A-MEM | 0.319 (↑4.2%) | 0.310 | 0.586 | 0.381 | 0.272 | 0.269 | 0.091 | 0.325 |
| EXPEL | 0.321 (↑4.9%) | 0.312 | 0.570 | 0.388 | 0.294 | **0.347** | 0.075 | 0.263 |
| Ours | **0.351 (↑14.6%)** | **0.335** | **0.596** | **0.393** | 0.299 | 0.347 | **0.099** | **0.391** |

Table 2: Performance comparison across seven QA datasets in training. Avg. column also reports relative improvement (%) compared to Search-R1 as the base. [†] indicates in-domain datasets, while [⋆] denotes out-of-domain datasets.

| Methods | Avg. (↑ / ↓ vs. Search-R1) | General QA | | | Multi-Hop QA | | | |
|---|---|---|---|---|---|---|---|---|
| | | NQ[⋆] | TriviaQA[⋆] | PopQA[⋆] | HotpotQA[†] | 2wiki[⋆] | Musique[⋆] | Bamboogle[⋆] |
| **Qwen3-8B** | | | | | | | | |
| Search-R1 | 0.395 (–) | 0.384 | 0.651 | 0.429 | **0.391** | 0.386 | 0.143 | 0.380 |
| Raw Trajectory | 0.400 (↑1.27%) | **0.406** | 0.657 | 0.433 | 0.376 | 0.367 | 0.139 | 0.423 |
| A-MEM | 0.403(↑2.03%) | 0.398 | 0.656 | **0.436** | 0.389 | **0.409** | 0.138 | 0.398 |
| EXPEL | 0.371 (↓6.08%) | 0.362 | 0.621 | 0.407 | 0.354 | 0.375 | 0.121 | 0.357 |
| Ours | **0.408 (↑3.29%)** | 0.386 | **0.662** | 0.434 | 0.387 | 0.403 | **0.152** | **0.435** |
| **Qwen3-4B** | | | | | | | | |
| Search-R1 | 0.375 (–) | 0.357 | 0.625 | 0.426 | 0.354 | 0.402 | 0.115 | 0.348 |
| Raw Trajectory | 0.415 (↑10.67%) | 0.403 | 0.624 | 0.434 | **0.420** | **0.428** | 0.186 | 0.412 |
| A-MEM | 0.388 (↑3.47%) | 0.393 | 0.603 | 0.439 | 0.385 | 0.322 | 0.157 | 0.418 |
| EXPEL | 0.337 (↓10.13%) | 0.322 | 0.577 | 0.399 | 0.311 | 0.363 | 0.081 | 0.305 |
| Ours | **0.426 (↑13.60%)** | **0.408** | **0.646** | **0.462** | 0.410 | 0.407 | **0.189** | **0.463** |

our memory workflow influences model performance in zero-training settings, i.e., during direct inference. **(2) Training Impact:** We investigate how the memory architecture affects RL training dynamics, focusing on convergence speed and the final performance achieved.

We compare our framework against a spectrum of methods ranging from standard prompting to advanced memory-augmented agents. We classify these baselines into three categories based on their utilization of experience:

- **Memory-Free Approaches: Direct Inference** relies solely on the internal parametric knowledge of the LLM. **Chain-of-Thought(CoT)** (Wei et al., 2022) enables large language models to solve complex problems by breaking them down into a series of intermediate reasoning steps before providing the final answer. **Tool-Integrated Reasoning(TIR)** (Chai et al., 2025)and **Search-R1** (Jin et al., 2025): agents augmented with search tools. While Search-R1 further learns an RL-based tool-use policy, both lack any mechanism to reuse experience across tasks.

- **Raw Experience Replay: Raw Trajectory** stores raw execution traces and retrieves the nearest trajectory as a few-shot demonstration. It uses unprocessed, unstructured experiences, offering no abstraction or learnable retrieval strategy.

- **Static Abstract Memory:** We compare against two representative structured memory systems: **A-MEM** (Xu et al., 2025)constructs a Zettelkasten-style memory graph by generating structured

notes for new experience, linking them to past memories through LLM-based semantic similarity checks. **Expel** (Zhao et al., 2024) distills experiences into a flat set of high-level textual insights. Crucially, while these methods abstract experience, rely on LLM-generated abstractions whose quality cannot be verified, making the stored memories potentially incomplete or misleading.

## 5.3 EXPERIMENT SETTING

From the HotpotQA training set, we sample 1,000 examples to construct the memory and an additional 5,000 examples for weight training. For each training query, we retrieve the top-k (k=5) similar historical queries, and the activation probability of a meta-cognition node is computed by propagating similarity scores through the graph.We sample N=2 distinct meta-cognitive strategies and conduct an independent counterfactual evaluation for each sampled strategies separately.

We conduct experiments with two model scales, **Qwen-3-4B** and **Qwen-3-8B**. For the retrieval component, we adopt the 2018 Wikipedia dump (Karpukhin et al., 2020) as the knowledge source and employ the **E5** (Wang et al., 2024) retriever. We report **Exact Match (EM)** as the primary evaluation metric.

## 5.4 MAIN RESULTS

**Experimental Analysis: Memory-Guided Inference**. The detailed inference results are summarized in Table 1. On the 8B-scale model, our method demonstrates strong competitiveness, achieving an average score of 0.365, which represents a notable **+9.3%** relative improvement over the TIR baseline and ranks first among all contenders. The advantages of our method become even more dramatic on the smaller Qwen3-4B model. It achieves a staggering **+14.6%** relative improvement in average performance over the TIR baseline, this significant performance improvement on a model with limited capacity suggests that our method effectively addresses its inherent deficiencies by providing a robust and structured reasoning framework.

A particularly noteworthy finding is that the memory component of our method was constructed exclusively using data from HotpotQA, the single in-domain dataset. Despite this, our method not only excels on HotpotQA but also achieves state-of-the-art or highly competitive performance across all out-of-domain datasets, including NQ, TriviaQA, PopQA, and 2wiki. This outcome is a strong testament to the remarkable generalization capability of our approach. It demonstrates that the reasoning structures learned from HotpotQA are not merely overfitted patterns.

**Experimental Analysis: Memory-Guided Reinforcement Learning**. We further evaluated our method by integrating it into RL training process. The detailed training results are in Table 2.

On the `Qwen3-8B` model, our method achieves the best average performance (0.408), improving upon `Search-R1` baseline by **3.29%**. This shows that our method provides additional benefits even after the model is already optimized with RL. The gains are most notable on challenging out-of-domain datasets like `TriviaQA` and `Bamboogle`, suggesting our memory helps the RL agent learn more general reasoning strategies that transfer well to new tasks.

On the smaller `Qwen3-4B` model, the results are even more impressive. Our method achieves a remarkable **13.60%** relative improvement over `Search-R1`. As seen in our inference experiments, the benefit of our method is especially pronounced on smaller models. Remarkably, our trained `Qwen3-4B` model (0.426) outperforms the baseline `Qwen3-8B` model (0.395), demonstrating a significant gain in efficiency.

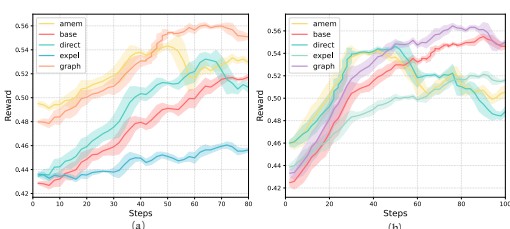

Figure 4: (a) Training curve of 4B models. (b) Training curve of 8B models.

In summary, adding our method to inference or RL training framework significantly boosts QA performance, especially for smaller models. Our structured memory helps the model learn general reasoning skills from the in-domain `HotpotQA` data and apply them successfully to other datasets. This allows smaller models to match or even exceed the performance of larger ones, offering a path to more efficient and capable models.

## 5.5 ABLATION STUDIES

We conduct ablation studies across three dimensions: (1) disabling memory weight updates (2) varying the number of meta-cognitions used as context and (3) altering the granularity of memory composition (i.e., API call structure).

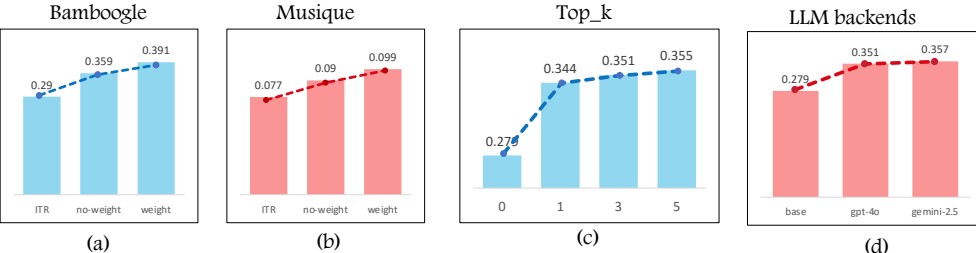

Figure 5: **Ablation studies of the structured memory framework.** (a) and (b) show the effect of disabling weight optimization. (c) varying the number of meta-cognition $k$. (d) generalization across LLM backends.

**Effect of Disabling Weight Optimization.** We first examine the impact of freezing the memory graph weights (i.e., no learning of edge confidence). In this setup, we keep all memory edges at uniform weight and retrieve strategies purely based on structural presence. As shown in figure 5(a)(b), performance drops significantly, particularly on `2WikiMultiHopQA`, indicating that learning to prioritize high-utility memory connections is crucial for effective strategy reuse. This validates our reinforcement-based update mechanism, which helps distinguish broadly useful meta-cognitions from less effective or overly specific ones.

**Varying the Number of Meta-Cognitions.** We further evaluate how the number of retrieved meta-cognitive strategies ($k$) affects model performance. Figure 5(c) presents the average accuracy of the 4B model on seven benchmarks as a function of the number of meta-cognitions. Increasing $k$ from 0 (no memory) to 3 leads to steady improvement, as more strategic signals are injected into the prompt. However, further increasing $k$ yields diminishing returns and can even introduce noise due to overlapping or irrelevant strategies. This highlights a trade-off between strategy diversity and clarity, and suggests that a moderate value of $k = 3$ offers the best balance between guidance and prompt efficiency. The detailed results are shown in Table 4.

**Generalization across LLMs backends.** To evaluate whether our memory construction is tied to a specific LLM API, we replace the original `OpenAI gpt-4o` model with `Gemini-2.5-pro` and rerun the downstream evaluation using the same memory graph. As shown in Table 5, our memory-augmented approach consistently outperforms its non-memory counterpart even under a different LLM backend, though the absolute numbers differ slightly due to model capability gaps. This demonstrates that our structured memory graph and retrieval-guided prompting strategy are largely *model-agnostic*, enabling plug-and-play use across modern foundation models.

## 6 CONCLUSION

In this paper, we address the dual challenges of inefficient decision-making and poor experience reuse in LLM-based agents. We introduce a trainable, multi-level graph memory framework that structurally encodes historical queries, policy trajectories, and high-level metacognitive strategies. This design facilitates explicit strategy recall and integrates memory into the RL loop to guide and accelerate policy optimization.

Unlike prior works that rely on either implicit optimization or static prompting, our approach unifies explicit memory with dynamic learning. By updating memory weights via RL signals, the framework selectively reinforces high-utility strategies and re-injects them into the agent's training process through prompt augmentation. This mechanism promotes strategic transfer and generalization from past experiences. Our experiments demonstrate that this method not only improves reasoning accuracy at inference time but also accelerates convergence during RL training, ultimately yielding superior final performance and strong generalization across diverse tasks.

## 7 ETHICS STATEMENT

This study uses publicly available datasets and does not involve private or confidential data. No human participants are included, and we ensure fairness and transparency in our model's design and deployment.

## 8 REPRODUCIBILITY STATEMENT

The model code, datasets, and experimental setup are available upon request. Detailed instructions for reproducing our experiments are provided to ensure transparency and facilitate further research.

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

## A    USE OF LLMs

In this study, LLMs are employed solely for text refinement and grammatical correction. Their role is restricted to enhancing the clarity, coherence, and linguistic accuracy of the written content, thereby ensuring compliance with academic standards, without affecting the research design, methodology, or decision-making.

## B    EXPERIMENTAL SETUP DETAILS

### B.1    MULTI-TURN TOOL-INTERGRATED QA

When tackling QA benchmarks, we observe that incorporating external knowledge retrieval plays a crucial role in enhancing answer accuracy. To this end, we utilize the 2018 Wikipedia dump as our external knowledge base and adopt the E5 retriever for efficient document retrieval. Within our framework, the entire retrieval process is encapsulated as a modular MCP tool, which the LLM can invoke autonomously based on its reasoning needs.

Specifically, the Qwen3 model follows a structured interaction protocol in which reasoning and planning are carried out between `<think>` and `</think>` tags,tool invocation commands are embedded within `<tool_call>` and `</tool_call>`, and the corresponding outputs are captured between `<tool_response>` and `</tool_response>`. The model engages in multiple rounds of tool interaction until it either generates a final answer enclosed in `<answer></answer>` or reaches the maximum limit of tool invocations, which is set to six in our experiments.

A concrete case study B.1 is presented in the following table to illustrate how this structured retrieval mechanism enables the model to access external knowledge and perform informed reasoning. And the table G.3presents a representative prompt designed to encourage agent to invoke tool.

---

**Case 1: Example Response with Tool-Integrated Reasoning**

response: `<think>` Okay, let's tackle this question. The user is asking which U.S. state was historically dominated by the Republican party from 1860 to 1932 and is represented by State Senator George D. Maziarz.

First, I need to figure out the state in question. The key points here are the time period (1860-1932) and the Republican dominance. I remember that during the 19th century, the Republican Party was a major political force, especially after the Civil War. States like Ohio, New York, and Pennsylvania were known for their Republican strongholds. However, I'm not entirely sure which one fits the specific time frame mentioned.

Next, the mention of State Senator George D. Maziarz. I need to check if this senator is from a state that fits the historical Republican dominance. But I don't recall specific information about George D. Maziarz. Maybe I should look up his state of origin. However, since I can't access external resources right now, I need to rely on my existing knowledge.
... `</think>`

```
<tool_call>
{"name": "search-query_rag",
"arguments": {"query": "George D. Maziarz state", "topk": 3}}
</tool_call>}
```

`<tool_response>` Execute the tool search-query_rag successed - The args are: 'query': 'George D. Maziarz state', 'topk': 3 - The result is:...... `</tool_response>`
`<answer>` New York `</answer>`

---

## B.2 EXPERIMENTAL SETTINGS

Table 3: Comparison of Agent Memory Architectures

| Method | Memory Structure | Retrieval Mechanism | Trainable? | Key Limitation |
|---|---|---|---|---|
| Direct Inference/ COT/ TIR | None | N/A | No | No experience reuse across tasks |
| Raw Trajectory | Raw Flat Trajectories | Embedding Similarity | No | No abstraction; noisy |
| A-MEM | Semantic Graph | Similarity + Traversal | No | Heuristic; task-agnostic structure |
| Expel | Flat Insights | Semantic Similarity | No | No structure; static memories |
| **Ours** | **FSM-based Hierarchical Graph** | **RL-learned Utility Weights** | **Yes** | – |

### B.2.1 DETAILED BASELINE SPECIFICATIONS

To provide a clearer understanding of the baselines used in our experiments, we offer extended descriptions of each method, including their core mechanisms, how they utilize memory or retrieval, and their inherent limitations. The table 3 presents some of the main differences among several types of memory using baselines.

**Direct Inference** represents the most basic setting, where the model generates answers purely based on its parametric knowledge from pretraining. It does not use any historical experience, external tools, or structured memory, and thus serves as an anchor point for evaluating the benefit of addi-

tional components. While simple and efficient, it lacks the ability to recall past mistakes or exploit cross-query information, making it limited for tasks requiring planning or adaptive strategies.

**Chain-of-Thought (CoT)** (Wei et al., 2022) enhances the basic model by encouraging it to articulate intermediate reasoning steps before producing an answer. This improves performance on tasks requiring logical decomposition. However, CoT still does not incorporate any past experience, and the reasoning steps themselves may be unstable or hallucinated, as they are freshly generated per query and not grounded in historical trajectories or external validation.

**Tool-Integrated Reasoning (TIR)** (Chai et al., 2025)equips the agent with external tool-use capabilities, allowing factual queries to be resolved by integrating retrieved evidence. Although this greatly improves factual correctness, the method still lacks any persistent memory, each query is resolved independently, and the agent does not reflect on past successes or failures. Moreover, the tool-use strategy is rule-based and not learned from reward.

**Search-R1** (Jin et al., 2025)adopts RL to optimize tool-use decisions, allowing the agent to learn better search strategies than TIR. Nevertheless, the agent lacks structured experience retention and cannot build reusable abstractions, making it limited for cumulative learning scenarios.

**Raw Trajectory** stores raw past execution traces and retrieves the most similar trajectories as demonstrations for new queries. This method leverages episodic experience more directly than the previous baselines, but its memory remains unstructured: the system stores full trajectories without abstraction, relevance assessment, or quality control.

**A-MEM** (Xu et al., 2025)constructs a Zettelkasten-style memory graph by generating structured notes for each new experience and linking them to past memories through LLM-based semantic similarity checks.

**ExpeL** (Zhao et al., 2024)abstracts experiences into concise textual insights distilled from past trajectories. This improves interpretability and reduces redundancy relative to raw trajectories. However, its memory is static: once distilled, these insights do not adapt to new tasks or feedback. The system cannot evaluate which memories are more useful, nor update their importance across training, leaving it unable to perform long-term strategic refinement.

### B.2.2 OTHER EXPERIMENT SETTINGS

We exclusively use the **HotpotQA** dataset, both for model optimization and for constructing memory during the memory formation process. Evaluation is then carried out on the test or validation sets of seven diverse datasets, enabling assessment of performance both within the training domain and in out-of-domain settings. We report **Exact Match (EM)** as the primary evaluation metric. And for memory construction in A-Mem (Xu et al., 2025), Expel (Zhao et al., 2024), and our proposed method, where a high-capability large language model is required, we utilized GPT-4o.

We conduct experiments on seven datasets, where HotpotQA is selected as the in-domain test set, while the remaining six datasets are used for out-of-domain evaluation.

During the RL training phase, we use the rest of the HotpotQA training set as the training corpus. We adopt a batch size of 512 with a micro batch size of 64, and the rollout sampling is performed with a temperature of 1.0. To accelerate the rollout process of the LLM, we deploy vLLM v1 with a tensor parallel size of 1.

Specifically for the GRPO algorithm, the number of rollout samples (n) is set to 8. All experiments are conducted on a cluster of 8 NVIDIA A100 GPUs.

## C MEMORY GRAPH CONSTRUCTION

The detailed descriptions of each type of node in the memory graph are as follows:

- **Query Layer** $\mathcal{Q}$: Each node $q_i \in \mathcal{Q}$ represents a specific task instance, such as a user-issued query. It encapsulates the entirety of an interaction, including the initial input, the agent's generated output, the complete execution trajectory, and a resultant outcome label (e.g., success or failure).

- **Transition Path Layer** $\mathcal{T}$: Each node $t_j \in \mathcal{T}$ corresponds to a standardized decision-making pathway. These pathways are grounded in a predefined FSM $\mathcal{S}$, representing a canonical sequence of the agent's states and actions. This layer abstracts away instance-specific details to reveal underlying behavioral patterns.

- **Meta-Cognition Layer** $\mathcal{M}$: Each node $m_k \in \mathcal{M}$ encodes a high-level, human-readable strategic principle. These principles are distilled from a comparative analysis of successful and failed transition paths, representing generalized heuristics for effective problem-solving.

The pseudo-code for the overall process of the graph, which is composed of the specific paths of LLM models, is as shown in the algorithm 1. And in order to better explain the concept and workflow of the memory graph, we illustrate how our framework processes a specific HotpotQA query:*"Are the bands The Blakes and Big Supreme from the same country?"* in Figure 3.

## C.1 MAPPING TRAJECTORIES INTO THE FINITE STATE MACHINE

The FSM serves as a structured cognitive map that regularizes highly variable CoT traces. Instead of relying on raw natural-language reasoning—which is verbose, stylistically inconsistent, and difficult to compare across tasks—the FSM abstracts an entire reasoning episode into a series of interpretable cognitive states. These states cover the complete lifecycle of decision-making, including intent parsing, uncertainty assessment, planning, tool execution, evidence analysis, and failure diagnosis. This abstraction removes surface-level linguistic noise and allows the system to analyze trajectories purely in terms of decision logic, the overall architecture of which is depicted in Figure 6.

A key requirement for accurate modeling of agent behavior is capturing how the agent reacts to external information. For this reason, the FSM contains an `InformationAnalysis` state that branches into three distinct cognitive responses: `KnowledgeAligned` (external evidence matches internal belief), `KnowledgeGap` (evidence provides new information), and `InternalConflict` (evidence contradicts prior parametric knowledge). Many reasoning failures—such as hallucination or incorrect synthesis—occur exactly at such conflict points. Including these branches allows the memory system to identify and aggregate structurally similar reasoning failures across tasks.

We use a powerful LLM to map the unstructured text into a sequence of predefined cognitive states (prompt in Table G.3). A concrete HotpotQA FSM example is shown in Table C.

---

**Case 2: An example of Finite State Machine**

**Illustrative Decision Path.** The following sequence illustrates a canonical decision path encoded within our framework:

$$\text{Start} \to \text{CorrectGoalEstablished} \to \text{KnowledgeUncertainGap}$$
$$\to \text{StrategyPlanning} \to \text{SequentialDependentPlanning}$$
$$\to \text{ToolExecution} \to \text{InformationAnalysis}$$
$$\to \text{KnowledgeAligned} \to \text{DecisionMaking}$$
$$\to \text{InsufficientInformation} \to \text{AssumptionBasedReasoning}$$
$$\to \text{AnswerGeneration} \to \text{WrongAnswer}$$
$$\to \text{DiagnosisHub} \to \text{InternalKnowledgeConflict} \to \text{End}$$

This path represents a chain of cognitive states traversed by the agent. It begins with goal establishment, proceeds through planning and execution, encounters a knowledge gap leading to flawed reasoning, and concludes with self-diagnosis. By encoding such trajectories as nodes in the transition path layer, the graph provides a structured and abstract representation of a complex reasoning process, which can be analyzed, compared, and learned from.

---

Overall, the FSM enables the system to preserve the essential decision dynamics of the agent, including how it processes uncertainty and conflicting evidence, while eliminating noisy linguistic variability, providing a reliable basis for downstream learning and strategy induction.

---

**Algorithm 1** Hierarchical Memory Graph Construction and Update

---

1: **Input:** Memory Graph $\mathcal{G}$, new query $q_i$, policy $\pi$, FSM $\mathcal{S}$, sample count $N$, similarity threshold $K$.
2: **Ensure:** Updated Memory Graph $\mathcal{G}'$.
3: **procedure** UPDATEMEMORYGRAPH($\mathcal{G}, q_i, \pi, \mathcal{S}, N, K$)
4:     $T_s \leftarrow \emptyset, T_f \leftarrow \emptyset$                               $\triangleright$ Initialize sets for successful and failed paths
5:     $\mathcal{G} \leftarrow$ AddNode($\mathcal{G}, q_i$)                             $\triangleright$ Add current query to the graph
6:     **for** $n = 1$ **to** $N$ **do**                         $\triangleright$ Sample N trajectories from the policy
7:         $\tau_n \leftarrow$ SampleRollout($\pi, q_i$)
8:         $t_n \leftarrow$ GroundTrajectoryToPath($\tau_n, \mathcal{S}$)     $\triangleright$ Map trajectory to a canonical FSM path
9:         $\mathcal{G} \leftarrow$ AddNode($\mathcal{G}, t_n$)
10:        $\mathcal{G} \leftarrow$ AddEdge($\mathcal{G}, q_i, t_n$)
11:        **if** IsSuccess($\tau_n$) **then**
12:            $T_s \leftarrow T_s \cup \{t_n\}$
13:        **else**
14:            $T_f \leftarrow T_f \cup \{t_n\}$
15:        **end if**
16:     **end for**
17:     $M_{new} \leftarrow$ InduceMetaCognition($q_i, T_s, T_f, \mathcal{G}, K$)     $\triangleright$ Derive new strategic principles
18:     **for each** new meta-cognition $m$ **in** $M_{new}$ **do**
19:         $m_{exist} \leftarrow$ FindMatchingMetaCognition($m, \mathcal{G}$)
20:        **if** $m_{exist}$ is **null then**
21:            $m_{final} \leftarrow$ CreateNewMetaCognitionNode($m$)
22:            $\mathcal{G} \leftarrow$ AddNode($\mathcal{G}, m_{final}$)
23:        **else**
24:            UpdateConfidence($m_{exist}$)
25:            $m_{final} \leftarrow m_{exist}$
26:        **end if**
27:        **for each** path $t$ that generated $m$ **do**     $\triangleright$ Link paths to the principles they support
28:            $\mathcal{G} \leftarrow$ AddEdge($\mathcal{G}, t, m_{final}$)
29:        **end for**
30:     **end for**
31:     **return** $\mathcal{G}$
32: **end procedure**

33: **procedure** INDUCEMETACOGNITION($q_i, T_s, T_f, \mathcal{G}, K$)
34:     **if** $T_s \neq \emptyset$ **and** $T_f \neq \emptyset$ **then**               $\triangleright$ **Case 1:** High-confidence induction
35:         $t_s \leftarrow$ SelectOne($T_s$), $t_f \leftarrow$ SelectOne($T_f$)
36:         $m \leftarrow$ ContrastPaths($t_s, t_f$)           $\triangleright$ e.g., find first diverging decision
37:         **return** $\{m\}$
38:     **else if** $T_s = \emptyset$ **and** $T_f \neq \emptyset$ **then**          $\triangleright$ **Case 2:** Speculative induction
39:         $M_{spec} \leftarrow \emptyset$
40:         $Q_{sim} \leftarrow$ FindSimilarQueries($q_i, \mathcal{G}, K$)         $\triangleright$ Based on embedding similarity
41:         **for each** similar query $q_j$ **in** $Q_{sim}$ **do**
42:            **for each** successful path $t_j$ of $q_j$ **do**
43:                $M_{spec} \leftarrow M_{spec} \cup$ GetMetaCognitionsFromPath($t_j, \mathcal{G}$)
44:            **end for**
45:         **end for**
46:         **return** $M_{spec}$
47:     **else**
48:         **return** $\emptyset$                    $\triangleright$ No new insights if only successes or no rollouts
49:     **end if**
50: **end procedure**

---

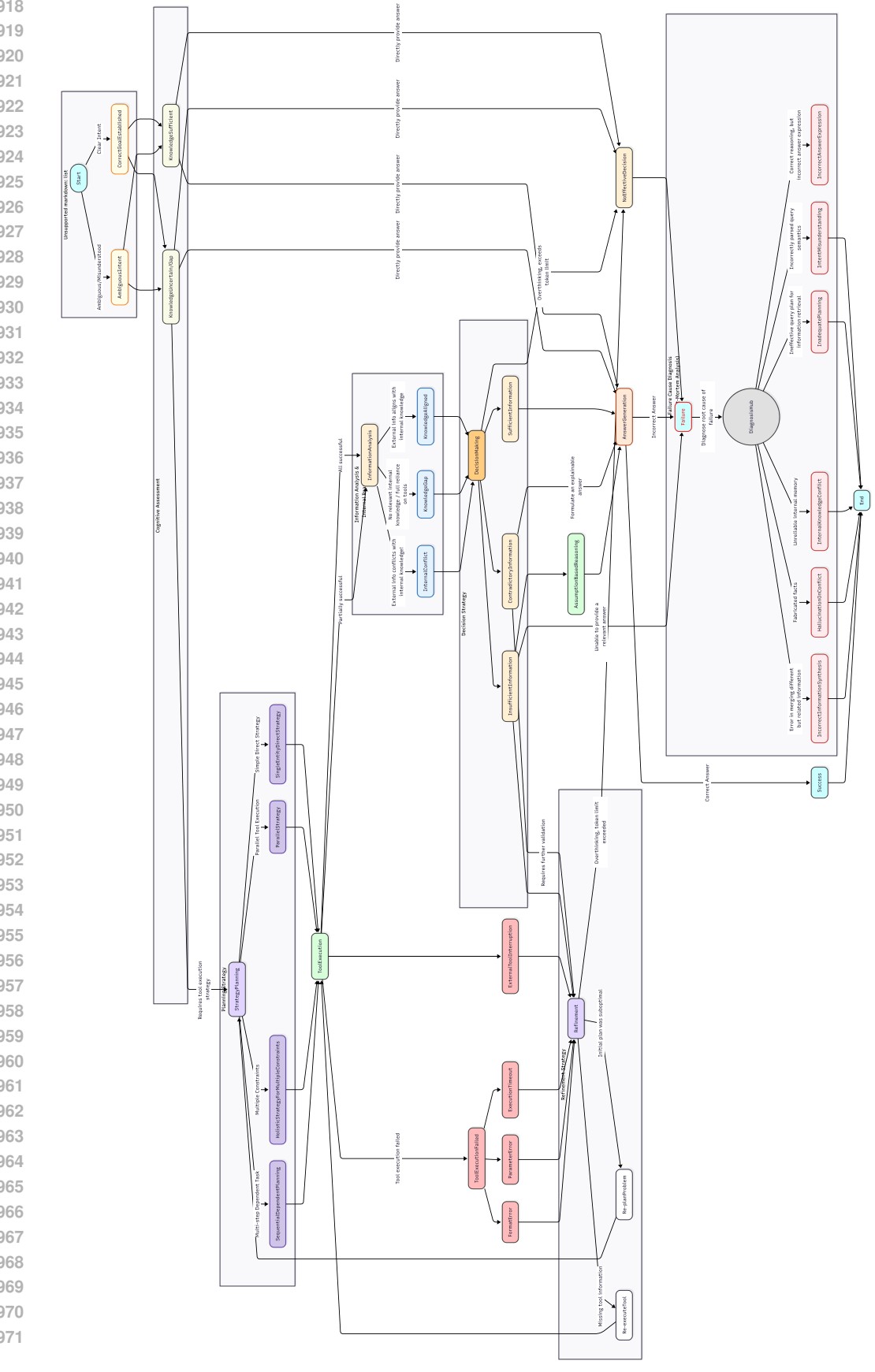

Figure 6: Finite State Machine

## C.2 META-COGNITION INDUCTION AND EVOLUTIONARY UPDATE

We define a **Meta-Cognition Node** ($m_k \in \mathcal{M}$) not as a summary of a specific trajectory, but as a *generalizable heuristic* distilled from FSM transition paths. While a transition path describes *what* the agent did, a meta-cognition captures *why* a behavior led to success or failure and provides explicit guidance for future decisions. For example: "When external evidence contradicts parametric memory, prioritize the external source unless the task demands internal knowledge." Thus, meta-cognitions convert raw trajectory patterns into reusable, high-level behavioral principles.

The meta-cognition layer is built and refined by a high-capacity *Teacher LLM* (e.g., GPT-4o), prompt is given in Appendix G.3. Given a new trajectory pair, the LLM performs an iterative **Knowledge Consolidation** process: it first abstracts and contrasts the FSM paths to extract a candidate insight; then checks whether this insight matches an existing principle; and finally either *refines* that principle by integrating new evidence or *creates* a new node when a previously unseen reasoning pattern is detected. This keeps the memory compact while ensuring that it evolves with accumulated experience.

**Three Meta-Cognition Induction Scenarios**   To cover diverse evidence conditions, meta-cognitions are induced under three complementary scenarios.

- **Scenario I: Intra-Query Causal Analysis (High Confidence).** When both a successful and failed trajectory exist for the *same* query, their FSM paths are aligned to locate the exact decision point of divergence. This yields the most reliable, strongly causal principles because all contextual factors are identical.
- **Scenario II: Inter-Query Analogical Analysis.** If a query contains only failures, we retrieve semantically similar queries with successful trajectories and compare their FSM patterns. Although less direct, this produces *analogical heuristics* that transfer successful behavioral patterns from related tasks.
- **Scenario III: Positive Behavior Distillation.** When only successful trajectories exist, the system extracts "good practice" heuristics such as effective planning or careful evidence verification, forming stable priors that guide reasoning in typical cases.

Together, these three scenarios allow the memory graph to capture causal rules, transferable strategies, and general positive behaviors, providing broad coverage of the agent's reasoning patterns.

## D THE IMPACT OF META-COGNITION

### D.1 QUALITATIVE ANALYSIS: THE CORRECTIVE ROLE OF META-COGNITION

To illustrate how meta-cognitive strategies regulate agent behavior, we present a case study in Figure 7 involving complex multi-hop reasoning.

In the **baseline scenario** (top), the agent relies on erroneous parametric memory, confidently hallucinating "Al Gore" as the 44th Vice President and explicitly refusing to use external tools (*"I don't think I need to use the search tool"*). This typifies intuition-driven failure.

In contrast, the **meta-cognition-augmented agent** (bottom) retrieves a strategy emphasizing *"structured planning and early synthesis verification."* Guided by this prior, the agent exhibits a distinct behavioral shift: it detects internal conflicts during reasoning (*"Wait, maybe I'm mixing up the numbers"*), suspends judgment to structure its thoughts, and strategically invokes the retrieval tool to verify the lineage. This case demonstrates that the induced meta-cognition effectively suppresses overconfident hallucinations and enforces a rigorous, evidence-based reasoning process.

### D.2 META-COGNITION ACTIVATION ANALYSIS ACROSS QUERY TYPES

To understand how the proposed memory graph enables task-adaptive reasoning, we analyze the activation probabilities of all 31 meta-cognition nodes under two representative multi-hop query

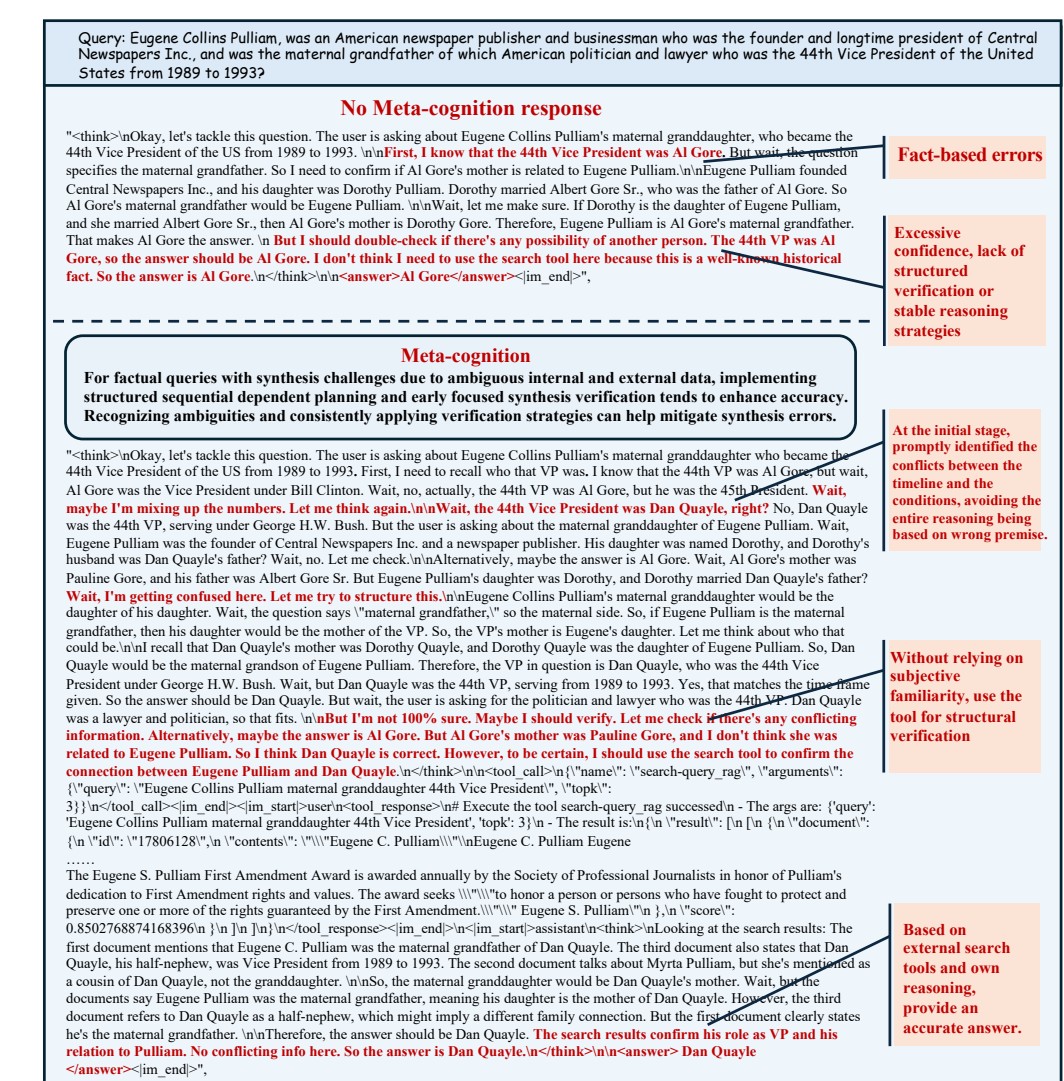

Figure 7: An example that demonstrates how meta-cognition affects reasoning

types—*Comparison* and *Bridge*. For each trained model, we record the selection probability of every meta-cognition node during inference and visualize this high-dimensional preference space using a t-SNE projection (Figure 8). Crucially, the purpose of this visualization is **not to perform semantic clustering**, but to characterize the **functional preference patterns** of the agent—specifically, which strategies are prioritized under distinct reasoning demands.

**General Meta-Cognitions.** We observe that a subset of meta-cognitions—such as *InternalKnowledgeCheck* exhibit consistently moderate activation across both Comparison and Bridge tasks. These meta-cognitions implement generic verification behaviors (e.g., quickly checking internal consistency or confirming tool outputs) and therefore serve as broadly applicable reasoning heuristics. Their stable appearance across task types indicates that the agent learns a set of **universally useful verification habits**.

**Task-Specific Meta-Cognitions.** In contrast, certain meta-cognitions—including *SequentialDependentPlanning* show sharply elevated activation in Bridge queries but low usage in Comparison queries. Bridge tasks typically rely on chained dependencies and frequently require the agent to identify missing information and execute follow-up tool calls. The selective activation of these strategies demonstrates that the memory system captures **scenario-specific reasoning preferences** rather than applying uniform heuristics.

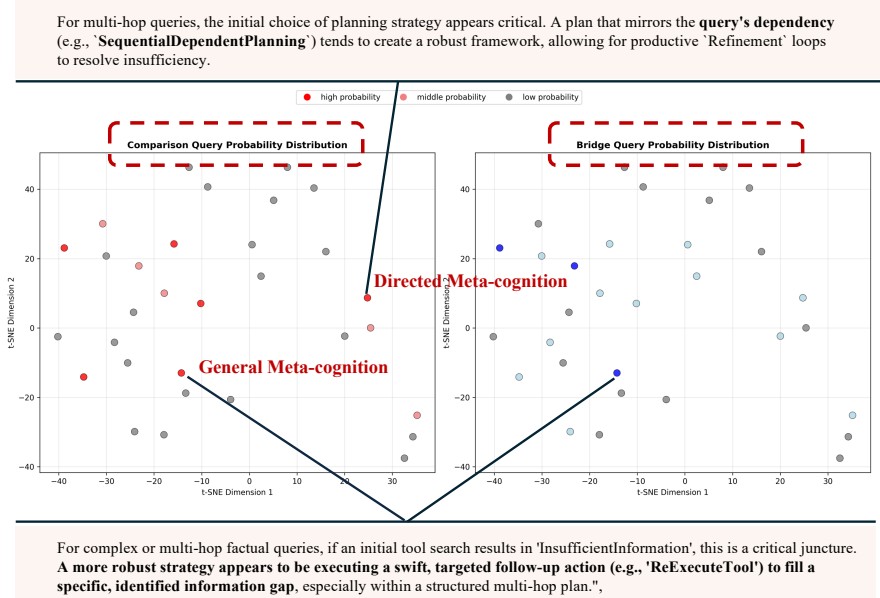

Figure 8: **Visualization of Meta-Cognitive Strategy Preferences.** The t-SNE projection maps the functional space of meta-cognition nodes based on their activation probabilities. This visualization reveals how the agent's preference shifts between *Comparison* and *Bridge* queries.

Taken together, these activation patterns show that the hierarchical memory graph does not merely store abstract advice—it supports **task-adaptive meta-cognition**. Comparison queries emphasize lightweight verification, while Bridge queries require multi-step planning or targeted gap resolution. Such differential behavior would not emerge from static rule-based memory systems; it arises specifically from our utility-weighted selection mechanism trained with reinforcement learning.

### D.3 EVOLUTION OF META-COGNITION SELECTION DURING RL TRAINING

To understand how RL reshapes the agent's strategic preferences, we tracked the selection probability of each meta-cognition node across three distinct training checkpoints (Batch 10, 150, and 250). Figure 9 visualizes this evolutionary process, providing a dynamic view of how the memory graph optimizes its retrieval distribution.

**Suppression of Low-Quality Meta-cognitions.** As illustrated in the top panel of Figure 9, the selection probability of brittle heuristics—such as `IncorrectInformationSynthesis` and `PrematureKnowledgeSufficient`—declines sharply throughout the training process $(0.136 \rightarrow 0.028 \rightarrow 0.017)$. These strategies typically encourage over-reliance on parametric memory or premature halting of the reasoning chain. Since trajectories guided by these heuristics frequently lead to incorrect answers (and thus negative rewards), the RL policy naturally suppresses their activation weights to maximize expected return.

**Emergence of High-Quality Meta-cognitions.** As shown in the top panel of Figure 9, the selection probability of several low-quality meta-cognitive heuristics (e.g., IncorrectInformationSynthesis, PrematureKnowledgeSufficient) drops sharply from $0.136 \rightarrow 0.028 \rightarrow 0.017$ over training. These heuristics typically encourage over-reliance on the agent's parametric memory or insufficient validation of dependency chains, leading to brittle multi-hop reasoning. RL naturally suppresses them because trajectories associated with these heuristics produce lower returns.

This analysis confirms that RL training does not merely tune the model's internal parameters—it fundamentally **reshapes the distribution over strategic principles**. The system actively filters out hallucination-prone shortcuts while cementing task-robust reasoning habits. This provides direct

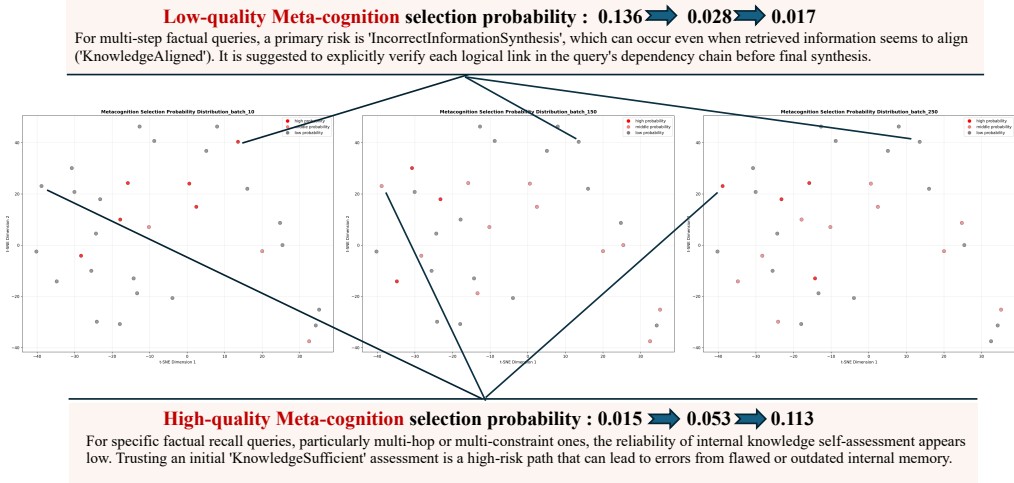

Figure 9: **Evolution of Strategic Preferences during RL Training.** We visualize the selection probability of meta-cognition nodes across three training checkpoints (Batch 10, 150, 250) using t-SNE projections. Each point represents a strategic principle; **color intensity** encodes the selection probability (Red = High, Gray = Low). As training progresses, the agent suppresses hallucination-prone heuristics (Top Panel) and converges toward robust verification strategies (Bottom Panel).

empirical evidence that our memory graph functions as a **trainable policy prior** that evolves from experience, rather than a static retrieval module.

# E EXPERIMENT ANALYSIS

## E.1 THE NUMBER OF THE META-COGNITION

To better understand how the quantity of retrieved meta-cognitive strategies affects agent performance, we evaluate four configurations: using 0 (no memory, denoted as TIR), 1, 3, and 5 strategies as contextual input. Results across seven QA benchmarks are presented in Table 4.

We observe that introducing even a single meta-cognitive strategy leads to a notable improvement over the baseline (TIR), especially on multi-hop tasks such as Bamboogle (+11.6%) and HotpotQA (+2.9%). This suggests that explicit strategic signals can substantially aid reasoning even in limited quantities. As the number of strategies increases, performance generally improves, but the marginal gains become smaller—likely due to redundancy or prompt saturation. The best overall result is achieved at $top\_k=5$, which balances diversity and relevance.

These findings imply that a moderate number of well-curated strategies can enhance generalization and decision quality, without incurring the risks of prompt overload or noise from irrelevant memories.

Table 4: Performance of different numbers of meta-cognition.

| Methods | General QA | | | Multi-Hop QA | | | | Avg. |
|---|---|---|---|---|---|---|---|---|
| | NQ* | TriviaQA* | PopQA* | HotpotQA[†] | 2wiki* | Musique* | Bamboogle* | |
| **Qwen3-4B** | | | | | | | | |
| TIR | 0.298 | 0.581 | 0.351 | 0.268 | 0.281 | 0.077 | 0.290 | 0.306 |
| $top_k = 1$ | 0.326 | 0.583 | 0.382 | 0.290 | 0.327 | 0.096 | 0.406 | 0.344 |
| $top_k = 3$ | 0.335 | 0.596 | 0.393 | 0.299 | 0.347 | 0.099 | 0.391 | 0.351 |
| $top_k = 5$ | 0.333 | 0.594 | 0.392 | 0.299 | 0.349 | 0.094 | 0.418 | 0.355 |

## E.2 CROSS-API MEMORY ROBUSTNESS

To further validate the portability and reliability of our structured memory graph, we construct the memory using two distinct LLM APIs: `gpt-4o` and `Gemini-2.5-pro`. These memory graphs are then integrated into the same downstream agent architecture (Qwen3-4B and Qwen3-8B), and evaluated across seven QA datasets. As shown in Table 5, the resulting performance differences are minor, with Gemini-based memory slightly outperforming its 4o counterpart in most cases.

Specifically, on the multi-hop benchmark Bamboogle, the Gemini-constructed memory shows a notable increase (e.g., +0.043 on Qwen3-8B), while maintaining parity or marginal gains in general QA datasets like TriviaQA and PopQA. These results indicate that while different APIs may introduce slight variations in strategy abstraction, our framework is robust to such differences and maintains high effectiveness regardless of the underlying model used to generate the memory.

Table 5: Performance comparison across LLM .

| Methods | General QA | | | Multi-Hop QA | | | | Avg. |
|---------|------|-----------|--------|-----------|--------|----------|------------|------|
|         | NQ* | TriviaQA* | PopQA* | HotpotQA[†] | 2wiki* | Musique* | Bamboogle* |      |
| **Qwen3-8B** | | | | | | | | |
| Ours(4o) | 0.316 | 0.622 | 0.382 | 0.358 | 0.354 | 0.128 | 0.392 | 0.365 |
| Ours(gemini) | 0.318 | 0.621 | 0.385 | 0.362 | 0.336 | 0.123 | 0.434 | 0.369 |
| **Qwen3-4B** | | | | | | | | |
| Ours(4o) | 0.335 | 0.596 | 0.393 | 0.299 | 0.347 | 0.099 | 0.391 | 0.351 |
| Ours(gemini) | 0.337 | 0.598 | 0.396 | 0.314 | 0.360 | 0.093 | 0.397 | 0.357 |

## E.3 ADDITIONAL STUDY: 2WIKI FOR MEMORY CONSTRUCTION

In our main experiments, we utilized HotpotQA to construct the FSM and memory graph due to its rich coverage of reasoning types (bridge, comparison, etc.). To rigorously examine whether our FSM abstraction relies on HotpotQA-specific structures, we conducted a cross-dataset portability study.

We repeated the entire memory construction pipeline using only trajectories sampled from the **2WikiMultiHopQA** dataset. The FSM definition (states and transitions) remained unchanged. We then evaluated this memory graph on the same seven downstream benchmarks using the Qwen3-4B model.

Table 6: Performance comparison across memory dataset.

| Methods | General QA | | | Multi-Hop QA | | | | Avg. |
|---------|------|-----------|--------|-----------|--------|----------|------------|------|
|         | NQ* | TriviaQA* | PopQA* | HotpotQA* | 2wiki[†] | Musique* | Bamboogle* |      |
| **Qwen3-4B** | | | | | | | | |
| TIR | 0.298 | 0.581 | 0.351 | 0.268 | 0.281 | 0.077 | 0.290 | 0.306 |
| 2wiki-memory | 0.317 | 0.591 | 0.394 | 0.290 | 0.308 | 0.091 | 0.322 | 0.330 |

As shown in the result (table 6), the memory graph constructed from 2Wiki still yields clear improvements over memory-free inference. This confirms that the FSM abstraction is dataset-agnostic and does not rely on Hotpot-specific structure. Our method is not tied to HotpotQA, and similar benefits are observed when using alternative multi-hop datasets.This experiment validates the generality and robustness of the proposed memory construction pipeline.

## F COMPUTATIONAL OVERHEAD ANALYSIS

To assess the practical efficiency of our framework, we provide a granular breakdown of the computational overhead, categorized into offline graph construction (Stage 1 & 2) and online inference

---

**Algorithm 2** Trainable Graph Weight Optimization via Policy Gradient

---

1: **Input:** Memory Graph $\mathcal{G}$ with initial weights $\mathbf{w}$, Training Queries $\mathcal{D}$, Agent model, Reward function $\mathcal{R}$, learning rate $\alpha$.
2: **Output:** Optimized Memory Graph $\mathcal{G}$ with updated weights $\mathbf{w}^*$.
3: **procedure** OPTIMIZEGRAPHWEIGHTS($\mathcal{G}, \mathcal{D}, \alpha$)
4:     **for each** query $q_{\text{new}}$ **in** $\mathcal{D}$ **do**
5:                                         ▷ — Step 1: Stochastic Guidance Selection —
6:         $m_k, p(m_k \mid q_{\text{new}}) \leftarrow$ SelectGuidingMetaCognition($\mathcal{G}, q_{\text{new}}$)
7:         **if** $m_k$ is **null then**                     ▷ No relevant guidance found
8:             **continue**
9:         **end if**
10:                                   ▷ — Step 2: Counterfactual Evaluation —
11:         $\text{Response}_{\text{with}} \leftarrow$ Agent.generate($q_{\text{new}}$, guidance $= m_k$)
12:         $R_{\text{with}} \leftarrow \mathcal{R}(\text{Response}_{\text{with}}, q_{\text{new}})$
13:         $\text{Response}_{\text{w/o}} \leftarrow$ Agent.generate($q_{\text{new}}$, guidance $=$ null)
14:         $R_{\text{w/o}} \leftarrow \mathcal{R}(\text{Response}_{\text{w/o}}, q_{\text{new}})$
15:         $\Delta R_k \leftarrow R_{\text{with}} - R_{\text{w/o}}$             ▷ Calculate reward gap (utility signal)
16:                                    ▷ — Step 3: Policy Gradient Update —
17:         $\nabla_{\mathbf{w}}\mathcal{L} \leftarrow -\Delta R_k \cdot \nabla_{\mathbf{w}} \log p(m_k \mid q_{\text{new}})$    ▷ Compute gradient for REINFORCE
18:         $\mathbf{w} \leftarrow \mathbf{w} - \alpha \cdot \nabla_{\mathbf{w}}\mathcal{L}$             ▷ Update all contributing weights
19:     **end for**
20:     **return** $\mathcal{G}$
21: **end procedure**

22: **procedure** SELECTGUIDINGMETACOGNITION($\mathcal{G}, q_{\text{new}}$)
23:                              ▷ Activate relevant subgraph based on semantic similarity
24:     $\mathcal{M}_{\text{act}} \leftarrow$ ActivateSubgraph($q_{\text{new}}, \mathcal{G}$)
25:     **if** $\mathcal{M}_{\text{act}}$ is empty **then**
26:         **return null**, 0
27:     **end if**
28:                            ▷ Compute relevance scores for all activated meta-cognitions
29:     **for all** $m \in \mathcal{M}_{\text{act}}$ **do**
30:         $S(m \mid q_{\text{new}}) \leftarrow 0$
31:         **for all** path $q_i \rightarrow t_j \rightarrow m$ in $\mathcal{G}$ **do**
32:             **if** $q_i$ is in activated subgraph **then**
33:                 $S(m \mid q_{\text{new}}) \leftarrow S(m \mid q_{\text{new}}) + \text{Sim}(q_{\text{new}}, q_i) \cdot w_{qt}^{(i,j)} \cdot w_{tm}^{(j,m)}$
34:             **end if**
35:         **end for**
36:     **end for**
37:                                ▷ Compute selection probabilities using softmax
38:     $Z \leftarrow \sum_{m' \in \mathcal{M}_{\text{act}}} \exp(S(m' \mid q_{\text{new}}))$
39:     **for all** $m \in \mathcal{M}_{\text{act}}$ **do**
40:         $p(m \mid q_{\text{new}}) \leftarrow \exp(S(m \mid q_{\text{new}}))/Z$
41:     **end for**
42:                        ▷ Stochastically sample a meta-cognition based on probabilities
43:     $m_k \leftarrow$ Sample($\mathcal{M}_{\text{act}}$, probabilities $= \{p(m \mid q_{\text{new}})\}$)
44:     **return** $m_k, p(m_k \mid q_{\text{new}})$
45: **end procedure**

---

latency (Stage 3). All measurements were conducted on the same hardware setup used for the main experiments.

## F.1 OFFLINE GRAPH CONSTRUCTION (ONE-TIME COST)

The construction of the memory graph is an offline preprocessing step performed once prior to RL training. As shown in Table 7, the primary costs stem from parsing trajectories into FSM paths and inducing meta-cognition via the powerful LLM.

- **Component-level Breakdown:** FSM parsing is efficient, averaging $4.086 \pm 0.879$ seconds per rollout. The meta-cognition update, which involves calling the powerful LLM (e.g., GPT-4o) to distill insights from trajectory pairs, requires $4.754 \pm 3.418$ seconds per query.

- **Total Preprocessing Time:** For a seed set of 500 queries, the cumulative time for Stage 1 is approximately 2.3 hours. The subsequent utility-weight optimization (Stage 2) is a batch process that takes an additional 2–3 hours.

Crucially, this combined cost of $\sim 5$ hours is **fully amortized**. The resulting graph structure and weights are frozen and reused throughout the entire RL training process and final inference, introducing zero recurrent cost during the agent's learning loop.

Table 7: **Offline Construction Cost Breakdown.** Statistics are measured over 500 seed queries. Note that Utility Update is a batch process measured end-to-end.

| Component | Unit | Time (Mean $\pm$ Std) | Variance | Notes |
|---|---|---|---|---|
| FSM Parsing | per rollout | $4.086 \pm 0.879$ s | 0.772 | Depends on trajectory length |
| Meta-cognition Induction | per query | $4.754 \pm 3.418$ s | 11.685 | Triggered once per 3 rollouts |
| **Stage 1 Total** | 500 queries | $\approx 2.3$ hours | – | One-time construction |
| **Stage 2 Total** | End-to-End | $\approx 2.0$–$3.0$ hours | – | One-time weight optimization |

### F.2 ONLINE INFERENCE LATENCY

A key concern is whether the longer prompts (augmented with retrieved meta-cognition) significantly slow down inference. To quantify this, we measured the runtime on the **HotpotQA test set** (using a batch size of 8 with the Qwen3-8B backbone).

As detailed in Table 8, the inclusion of meta-cognition increases the average batch processing time from 32.156s to 32.511s. This results in an additional latency of only **0.355 seconds per batch** (or $\sim 0.044$s per query). This represents a marginal overhead of approximately **1.1%**, which is negligible compared to the natural variability of LLM token generation. The minimum and maximum latencies are also nearly identical, confirming that our method does not introduce significant variance or instability to the inference pipeline.

Table 8: **Inference Time Comparison.** Statistics are derived from HotpotQA test dataset with a batch size of 8. The overhead introduced by meta-cognitive prompting is marginal ($\sim 1.1\%$).

| Setting | Unit | Mean Time (s) | Std (s) | Min (s) | Max (s) | Relative Overhead |
|---|---|---|---|---|---|---|
| w/o Meta-cognition | per batch | 32.156 | 3.759 | 26.715 | 38.675 | – |
| w/ Meta-cognition | per batch | 32.511 | 4.363 | 26.628 | 40.314 | $+1.10\%$ |
| **Difference** | **per batch** | **+0.355 s** | – | – | – | – |
| **Difference** | **per query** | **+0.044 s** | – | – | – | – |

## G PROMPT TEMPLATES

### G.1 MULTI-TURN TOOL-INTERGRATED QA PROMPT

When LLMs needs to interact with tools multiple times to answer a question, it is necessary to guide the LLM on which tools to use and how to use them. The specific prompt is shown in table G.3.

### G.2 CONSTRUCTING FSM PATH

Given the specific response path of the LLM and the complete structure of the state machine, we employ an LLM (e.g., GPT-4o) to map the generated answer onto one of the predefined paths in the state machine. The following shows the exact prompt used in G.3.

### G.3  META-COGNITION CONSTRUCTING

With both successful and failed state-machine paths available, we derive high-level meta-cognitions by contrasting the two. The following prompt G.3 illustrates how a pair of successful and failed paths under the same query is used to induce meta-cognition.

---

**Prompt A: System and User Prompt**

**SYSTEM PROMPT:**

```
# Tools
You may call one or more functions to assist with the user query.
You are provided with function signatures within <tools></tools> XML
    tags:
<tools>
{
  "name": "search-query_rag",
  "description": "MCP RAG Query Tool (Synchronous Version)
  Args:
    query: query text
    topk: The default number of documents returned is 3
  Returns:
    str: The formatted query result",
  "parameters": {
    "type": "object",
    "properties": {
      "query": {"title": "Query", "type": "string"},
      "topk": {"default": 3, "title": "Topk", "type": "integer"}
    },
    "required": ["query"]
  }
}
</tools>

# Tool call format
For each function call, return a JSON object with function name and
    arguments within <tool_call></tool_call> XML tags:
<tool_call>
{
  "name": <function-name>,
  "arguments": <args-json-object>
}
</tool_call>
```

**USER PROMPT:**

```
Answer the given question. After reasoning, if you find you lack
    some knowledge, you can call the search tool.
You may search as many times as you want.
If you find no further external knowledge is needed, you can
    directly provide the answer inside <answer> and </answer>,
    without detailed illustrations.
For example: <answer> Beijing </answer>.

Question: Which US State, historically dominated by the Republican
    party from 1860 to 1932, is represented by State Senator George
    D. Maziarz?
```

---

1404
1405
1406
1407
1408
1409
1410
1411
1412
1413
1414
1415
1416
1417
1418
1419
1420
1421
1422
1423
1424
1425
1426
1427
1428
1429
1430
1431
1432
1433
1434
1435
1436
1437
1438
1439
1440
1441
1442
1443
1444
1445
1446
1447
1448
1449
1450
1451
1452
1453
1454
1455
1456
1457

**Prompt B: Prompt for constructing FSM path**

**Instruction:** You are a metacognition analysis expert specialized in extracting *generalized decision principles and guidance strategies* from state machine execution paths.
**State machine transition rules:** {transitions_info}
**Core Requirements:**

1. **Generalizability Focus**: Output strategies and principles must be general, applicable to similar problems, without specific query details.

2. **Direct Usability**: Generated content should be directly usable as guidance principles for new problems.

3. **Principled Expression**: Use cautious guidance terms like "consider", "may help", "tends to" rather than definitive statements.

4. **Concise Effectiveness**: Output only the most core insights, avoid redundancy and complexity.

5. **Quality Control**: Strictly evaluate whether there is sufficient evidence to support new metacognition.

6. **Knowledge Confidence Awareness**: Recognize that LLM's internal knowledge confidence varies across queries — success patterns may be domain-specific.

7. **Uncertainty Acknowledgment**: Express appropriate uncertainty in guidance principles, avoiding overly definitive conclusions.

8. **Quantity Management**: When metacognition count exceeds 30, prioritize updating low-confidence existing metacognitions.

**Output Format (Quantity-Aware):**
Your output must be a JSON object with the following structure:

```
{
  "decision": "update" or "create" or "skip",
  "target_meta_id": <ID of metacognition to update (only when
      decision is "update")>,
  "reasoning": "Brief explanation including quantity management when
      count > 30.",
  "meta_cognition": {
    "summary": "Concise general guidance summary (use cautious
        language).",
    "strategy_principles": [
      {
        "principle": "...",
        "confidence": "high" | "medium" | "low",
        "confidence_score": 30 – 85
      },
      ...
    ],
    "overall_confidence": "high" | "medium" | "low",
    "evidence_paths": <int>,
    "uncertainty_note": "Brief acknowledgment of limitations or
        knowledge-dependency concerns."
  }
}
```

---

**Prompt C: Metacognition Prompt Specification**

**State machine transition rules:** {`transitions_info`}

**Core Requirements:**

1. **Generalizability Focus:** Output strategies and principles must be general, applicable to similar problems, without specific query details.

2. **Direct Usability:** Generated content should be directly usable as guidance principles for new problems.

3. **Principled Expression:** Use cautious guidance terms like "consider", "may help", "tends to" rather than definitive statements.

4. **Concise Effectiveness:** Output only the most core insights, avoid redundancy and complexity.

5. **Quality Control:** Strictly evaluate whether there is sufficient evidence to support new metacognition.

6. **Knowledge Confidence Awareness: Recognize that LLM's internal knowledge confidence varies across queries—success patterns may be domain-specific.**

7. **Uncertainty Acknowledgment: Express appropriate uncertainty in guidance principles, avoiding overly definitive conclusions.**

8. **Quantity Management:** When metacognition count exceeds 30, prioritize updating low-confidence existing metacognitions.

**Critical Self-Reflection Requirements:**

- **Pattern Validity:** Question whether identified patterns truly represent generalizable principles.

- **Knowledge Dependency: Consider if success stems from strategy effectiveness or the LLM's domain familiarity.**

- **Evidence Sufficiency:** Demand higher evidence standards for strategies that could mislead future queries.

- **Simplicity Over Complexity:** Favor simple, robust principles over complex, brittle ones.

**Metacognition Quantity Control Strategy:**

**When metacognition count $\leq$ 30:**  • Normal decision making: create, update, or skip based on evidence quality.
- Prefer creating new metacognition when patterns are sufficiently distinct.
- **Express appropriate uncertainty in new metacognitions.**

**When metacognition count $>$ 30:**  • **Strongly prefer UPDATE over CREATE**: Prioritize improving existing low-confidence metacognitions.
- Only create new metacognition if the pattern is exceptionally valuable and completely distinct.
- Target metacognitions with confidence levels "low" or "medium" for updates.

**Analysis Focus:**

1. **Success Pattern Identification:** Abstract reusable decision patterns from successful paths.

2. **Failure Cause Summary:** Identify generalizable errors to avoid from failed paths.

3. **State Transition Optimization:** Extract best practice principles for state machine execution.

4. **Knowledge Dependency Assessment: Evaluate whether patterns might be specific to certain knowledge domains.**

5. **Existing Knowledge Enhancement:** When quantity is high, focus on strengthening weak metacognitions.

**Decision Options:**

- **create:** Create new metacognition (when discovering valuable and distinct patterns, or when quantity $\leq 30$).

- **update:** Update existing metacognition (preferred when quantity $> 30$, especially targeting low-confidence ones).

- **skip:** Skip metacognition operation (when evidence is insufficient or has no new value).

**Skip Metacognition Situations:**

- Path data quality is poor, patterns are unclear.

- Existing metacognition already covers the pattern, new evidence shows no significant improvement.

- Success/failure path differences are not obvious, difficult to extract effective strategies.

- **Cannot distinguish whether success stems from strategy effectiveness or knowledge domain familiarity.**

- When quantity $> 30$ and no suitable low-confidence metacognition found for update.

**Output Format (Quantity-Aware):** Your output must be a JSON object containing:

```
{
  "decision": "update" or "create" or "skip",
  "target_meta_id": (when decision is update) ID of metacognition to
      update,
  "reasoning": "Brief decision analysis, must include quantity
      management when count > 30.",
  "meta_cognition": {
    "summary": "...",
    "strategy_principles": [
      {"principle": "...", "confidence": "high", "confidence_score":
          80},
      {"principle": "...", "confidence": "medium", "confidence_score
          ": 60}
    ],
    "overall_confidence": "medium",
    "evidence_paths": 7,
    "uncertainty_note": "..."
  }
}
```

