# OpenReview forum: "From Experience to Strategy: Empowering LLM Agents with Trainable Graph Memory"
_ICLR.cc/2026/Conference — ICLR 2026 Conference Desk Rejected Submission_

### Official Review · Reviewer_scsu · 2025-10-17

**Soundness:** 2
**Presentation:** 1
**Contribution:** 2
**Rating:** 4
**Confidence:** 3

**Summary:**

This paper explores the use of dynamic, structured explicit memory to guide and enhance the policy learning. The authors propose a method where a graph is constructed to represent logical connections between user queries, decision states, and higher-level strategic cognition. An RL-based approach is then designed to dynamically optimize the graph’s weights. Finally, the framework jointly optimizes both the graph weights and the policy in an end-to-end RL training pipeline, enabling effective utilization of prior experiences and enhancing the policy’s reasoning capabilities.

**Strengths:**

1. The proposed approach introduces a novel way of leveraging a graph structure to represent logical connections between user queries, intermediate states, and abstracted experiences. This structured memory facilitates reasoning and strategic decision-making.

2. The framework enables both memory graph weights and the policy model to be jointly optimized in an end-to-end manner using RL. This ensures that the memory graph is dynamically updated to reflect the latest logical flow, while the policy can effectively utilize the structured experiences to enhance its reasoning capabilities.

3. The experiments provide promising results, demonstrating the effectiveness of the proposed approach in leveraging explicit memory for policy learning.

**Weaknesses:**

1. **Limited Discussion of Related Work**

    The paper lacks a detailed comparison with prior works, such as Expel and G-Memory.
   * Compared to Expel, which also explicitly abstracts experiences and insights from past trials, the differences seem to lie in both the model's parametric updates and the structured representation of experiences.
   * For G-Memory, both approaches utilize graph structures for memory representation, but the distinctions between these methods are not clearly articulated. A more thorough discussion of these works and how they compare to the proposed framework would strengthen the paper.

2. **Clarity and Accessibility**

    The overall method is difficult to understand due to overly abstract descriptions and unclear explanations. While the memory graph is the core contribution, its component, such as the "Transition Path Layer," and "Meta-Cognition Layer", are too abstract, even with the additional details provided in the Appendix. The lack of intuitive examples makes it challenging to grasp key concepts, such as what constitutes a transition path or a state, and what qualifies as meta-cognition? Similarly, the visualization of the graph is vague and lacks sufficient detail. Important notations, such as "ITR," which frequently appears in the main text, are not explicitly defined. Providing more intuitive examples and clearer writing would make the paper significantly more accessible.

3. **Constrained Experiments**

    The experimental setup is limited and does not fully align with the scenarios where the proposed approach could be most beneficial. Multi-hop QA (MHQA) tasks may not be the most suitable benchmark for evaluating how agents can benefit from past experiences. Other benchmarks that emphasize strategic planning and prior knowledge, such as WebShop, ALFWorld, or GAIA, BrowseComp, may better align with the paper's motivation. The MHQA tasks themselves are relatively simple, as most queries can be resolved with just 2–3 tool calls. The authors further limit the maximum number of tool calls to fewer than 10, which makes the tasks even less demanding and reduces the need for strategic planning. Alternatively, more challenging benchmarks, such as BrowseComp, could better demonstrate the proposed framework’s utility. Even within the MHQA tasks, the paper lacks illustrative examples showing how the retrieved subgraph contributes to answering queries. Including such examples would better demonstrate the practical effectiveness of the memory graph.

**Questions:**

Please refer to the Weakness part

---

> ### Author Response · Authors · 2025-11-22
> **Response to Reviewer scsu (part 1)**
>
> We sincerely thank the reviewer for the thoughtful feedback. We are encouraged by your recognition of our **graph-based structure** and the **end-to-end joint optimization** framework.
>
> Your constructive criticism regarding the **comparison with prior work**, the **presentation clarity**, and the **experimental scope** has helped us identify key areas for improvement. In the revised manuscript, we have significantly expanded the Related Work section, integrated concrete visualizations into the main text, and added case studies to demonstrate utility.
>
> Below, we address your specific concerns point-by-point.
>
>
> ### **Q1: Comparison with other baseline and G-Memory**
>
> Thank you for pointing out the need for clearer and more explicit baseline definitions. We agree that differentiating baselines along the axes you listed—memory construction, retrieval mechanism, trainability, and training-time vs inference-only usage—is essential for understanding the contribution of our method.
>
> In the revised **Section 5.2**, we have reorganized the baselines into three coherent categories and provided concise, main-text definitions (not limited to the appendix). The updated descriptions now read as follows:
>
> **1. Memory-Free Approaches (The Lower Bounds)**
> These methods establish the baseline performance without cross-task experience.
> *   **Direct Inference:** represents the most basic setting, where the model generates answers purely based on its parametric knowledge from pretraining. It does not use any historical experience, external tools, or structured memory, and thus serves as an anchor point for evaluating the benefit of additional components.
> *   **CoT (Chain-of-Thought):** enhances the basic model by encouraging it to articulate intermediate reasoning steps before producing an answer. This improves performance on tasks requiring logical decomposition.
> *   **TIR (Tool-Integrated Reasoning) & Search-R1:** Agents augmented with search tools (and RL-based tool policies in Search-R1). Crucially, while they can access external *world knowledge* (via tools), they lack any mechanism to accumulate *experiential knowledge* across tasks.
>
> **2. Raw Experience Replay (Unstructured Memory)**
> *   **Raw Trajectory:** Represents a standard RAG approach that stores raw execution traces. It retrieves the nearest trajectory as a few-shot demonstration. Its key limitation is the lack of abstraction—it uses unprocessed, noisy experiences without any learnable strategy.
>
> **3. Static Abstract Memory**
> We compare against representative structured systems:
> *   **A-MEM:** constructs a Zettelkasten-style memory graph by generating structured notes for each new experience and linking them to past memories through LLM-based semantic similarity checks.
> *   **Expel:** abstracts experiences into concise textual insights distilled from past trajectories.
> *   **G-memory:** A three-tier graph structure was designed to manage the collective memory of the multi agent system.
> *   **Critical Distinction:** As clarified in the revised text, the memory is static: once distilled, these insights do not adapt to new tasks or feedback. The system cannot evaluate which memories are more useful, nor update their importance across training, leaving it unable to perform long-term strategic refinement.
>
>
> **Summary of Differences**
> To directly address your request for a comparison along specific axes, we have added **Table 3** to the main text. This demonstrates that our method is the only one that combines a **hierarchical structure** with **RL-based trainability**.
>
> | Method | Memory Structure | Retrieval Mechanism | Trainable? | Key Limitation |
> | :--- | :--- | :--- | :--- | :--- |
> | **Direct Inference/COT/ITR** | None | N/A | No | No experience reuse |
> | **Raw Trajectory** | Raw Flat Trajectories | Embedding Similarity | No | No abstraction; noisy |
> | **A-MEM** | Semantic Graph | Similarity + Traversal | No | Heuristic; task-agnostic |
> | **Expel** | Flat Insights | Semantic Similarity | No | Static; no reward alignment |
> | **G-memory** | Hierarchical Graph | Bi-directional Traversal | No | Static; no reward alignment |
> | **Ours** | **FSM-based Hierarchical Graph** | **RL-learned Utility Weights** | **Yes** | -- |
>
> These clarifications ensure that the performance gains reported in our results can be correctly attributed to the **trainable and structured nature** of our memory, rather than simple tool access or raw retrieval.

---

> ### Author Response · Authors · 2025-11-22
> **Response to Reviewer scsu (part 2)**
>
> ### **Q2: Clarity, Abstraction, and Visualization**
>
> We fully acknowledge the reviewer’s concern that the method description was previously too abstract. In response, we have substantially strengthened the clarity of the presentation. Specifically, beyond adding the concrete end-to-end example in **Figure 3**, we have also rewritten **Section 4.1** to provide clearer explanations of the key concepts within the limited space of the main text, and expanded **Appendix C** with detailed, step-by-step descriptions of the FSM construction and meta-cognition induction process.
>
> For example a complete flowchart (**Figure 3**) shows:
> *   **Raw Trace:** An agent explicitly failing due to hallucination.
> *   **Transition Path:** How this trace maps to states: `ToolExecution` to `InternalConflict` to `DiagnosisHub`.（The FSM is a structured way to convert messy, inconsistent Chain-of-Thought text into a clean sequence of cognitive states that represent what the agent is actually doing during reasoning by powerful LLM.）
> *   **Meta-Cognition:** The resulting strategic text: *"Prioritize external verification when internal conflict arises."* (A meta-cognition is a high-level strategic rule distilled from multiple transition paths, describing why certain behaviors succeed or fail.)
>
> This example concretely defines what a "state" and "path" look like in practice.
>
>
> ### **Q3: Experimental Scope and Suitability (New ALFWorld Experiment)**
>
> While we believe multi-hop QA rigorously tests internal reasoning logic and hallucination correction (as noted in our paper), we truly appreciate your valuable suggestion to evaluate our framework in a decision-heavy setting.
>
> **1. Justification for MHQA**
> *   **Cognitive Density:** MHQA tasks like HotpotQA and 2Wiki require intense **internal reasoning**—resolving bridge entities, comparing attributes, and synthesizing conflicting evidence. This perfectly targets the "Meta-Cognitive" layer (planning & verification) that our graph is designed to improve.
> *   **Hallucination Sensitivity:** MHQA is highly sensitive to parametric hallucinations (as seen in our Al Gore case). This makes it an ideal testbed to verify if our memory graph can enforce "evidence-based reasoning" over "intuition," which is harder to isolate in environment-interaction tasks.
>
> *   **Complexity of "2-3 Tool Calls"**
> While the *number* of tool calls is low, the *decision logic* is complex. A successful HotpotQA trajectory often involves multiple cognitive checks (e.g., `InternalKnowledgeCheck`, `RefineAfterGapDetection`) that are invisible in the step count but crucial for accuracy. Our FSM reveals this hidden complexity.
>
> *   **Illustrative Case Study (Sub-graph Utility)**
> To address your request for examples of "how the retrieved subgraph contributes," we added **Figure 7** (Case Study) in the Appendix D.
> Guided by this meta-cognition, the agent triggers a self-correction loop (*"Wait, maybe I'm mixing up the numbers"*), explicitly calls the search tool, and corrects the answer to Dan Quayle.
> This directly demonstrates the practical utility of the memory graph in correcting reasoning failures.
>
> **2. New Experiment: Generalization to ALFWorld**
> To verify the portability of our method to embodied decision-making, we conducted an additional experiment on the **ALFWorld** benchmark using **GPT-3.5-turbo** as the backbone.
> *   **FSM Adaptation:** Although our original FSM was designed for tool-use QA, the underlying cognitive dynamics (Planning $\to$ Action $\to$ Observation $\to$ Refinement) are isomorphic to embodied tasks. We utilized the "flexible adaptation" prompt, allowing the LLM to map ALFWorld's specific actions into abstract paths based onour generalized cognitive states.
> *   **Baseline:** We compared our method against **Expel**, a strong static memory baseline that also abstracts insights.
>
> **Results and Analysis**
> As shown in the table below, our method achieves a **Success Rate of 80.60%**, outperforming Expel (75.37%).
>
> | Task Type | Expel (Success %) | Ours (Success %) | Improvement |
> | :--- | :--- | :--- | :--- |
> | **Pick & Place** | 67.0 | **79.0** | +12.0% |
> | **Pick Two & Place** | 71.0 | **76.0** | +5.0% |
> | **Clean & Place** | 87.0 | **90.0** | +3.0% |
> | **Heat & Place** | 78.0 | 74.0 | -4.0% |
> | **Cool & Place** | 95.0 | 90.0 | -5.0% |
> | **Look At Object** | 44.0 | **67.0** | +23.0% |
> | **Average** | **75.37%** | **80.60%** | **+5.23%** |
>
> **3. Conclusion**
> Our method yields substantial gains in challenging sub-tasks like `Look At Object` (+23%) and `Pick & Place` (+12%). This confirms **Universality of FSM:** The graph-based cognitive abstraction is not limited to QA tools but successfully transfers to sequential decision-making environments.
>
> We believe these additions make the paper significantly more accessible and clearly justify the experimental choices.

---

### Official Review · Reviewer_z3UM · 2025-10-27

**Soundness:** 2
**Presentation:** 3
**Contribution:** 3
**Rating:** 6
**Confidence:** 3

**Summary:**

The paper introduces a trainable, multi-layered graph memory framework designed to enhance the reasoning and adaptability of large language models. This structured memory addresses the challenges of limited interpretability and catastrophic forgetting in prior approaches. It represents an agent’s past experiences as structured decision paths, which are distilled into high-level, human-interpretable meta-cognitive strategies. Empirical results demonstrate the framework’s effectiveness and support its claims.

**Strengths:**

1. The framework is clearly described, making it easy to understand both its overall design and the role of each component. The key methods are thoroughly explained and supported with equations.

2. The experiments are well designed and make use of state-of-the-art models. The selected tasks are thoughtfully chosen, providing broad and representative coverage. The ablation study effectively demonstrates the contribution of each component.

**Weaknesses:**

1. The paper lacks an analysis of time consumption. Although the proposed method performs well as reported, there is no discussion of the additional computational overhead it may introduce. The method could potentially involve significant computational costs, which might limit its practical adoption compared to more efficient alternatives.

2. The paper claims that employing a structured graph memory can “distill agent trajectories into high-level, human-interpretable strategic meta-cognition.” However, there is limited evidence showing how the graph (queries, transition paths, and meta-cognitions) evolves during training. Including such an analysis would strengthen the paper’s soundness and provide deeper support for the empirical results.

**Questions:**

My questions are as follows, based on the weaknesses identified above:

1. **Time analysis**: Could the authors provide a detailed runtime analysis—ideally broken down by component (e.g., graph construction, meta-cognition induction, and update)—to demonstrate that the proposed method achieves performance improvements without compromising practicality?

2. **Graph visualization**: Is it possible to visualize the memory graph at different training stages to help readers better understand how the proposed method enhances interpretability and adaptability?

---

> ### Author Response · Authors · 2025-11-22
> **Response to Reviewer z3UM (part 1)**
>
> We sincerely thank the reviewer for the positive assessment and for recognizing the **clarity of our framework design** and the **comprehensiveness of our experiments**. We are particularly encouraged that you found the ablation studies effective and the task selection representative.
>
> Your constructive feedback regarding **computational overhead** and **graph evolution** points to two critical aspects of practicality and interpretability. We have significantly expanded our analysis in the revised manuscript.
>
> Below, we provide detailed responses to your specific questions.
>
> ### **Q1: Detailed Runtime Analysis**
>
> We thank the reviewer for requesting a granular breakdown of the runtime costs. We agree that assessing the practicality requires distinguishing between construction costs and inference latency. In the revised **Appendix F**, we have added a detailed time profiling analysis.
>
> **1. Offline Construction (One-Time Cost)**
> The FSM construction and weight optimization are strictly **offline processes** performed once prior to RL training.
> *   **FSM Parsing:** This step is efficient, averaging **4.09s** per trajectory.
> *   **Meta-cognition Induction:** Using the Teacher LLM takes **~4.75s** per query.
> *   **Total Time:** For a seed set of 500 queries, Stage 1 takes $\approx$ 2.3 hours, and the Stage 2 utility optimization takes an additional 2–3 hours.
> Crucially, this combined cost of $\sim$5 hours is **fully amortized**. The resulting graph structure and weights are frozen and reused throughout the entire RL training and inference phase, introducing **zero recurrent cost** for graph maintenance during the agent's learning loop.
>
> | Component | Unit | Time (Mean ± Std) | Notes |
> | :--- | :--- | :--- | :--- |
> | **FSM Parsing** | per rollout | 4.086 ± 0.879 s | Depends on trajectory length |
> | **Meta-cognition Induction** | per query | 4.754 ± 3.418 s | Triggered once per 3 rollouts |
> | **Stage 1 Total** | 500 queries | $\approx$ 2.3 hours | One-time construction |
> | **Stage 2 Total** | End-to-End | $\approx$ 2.0–3.0 hours | One-time weight optimization |
>
> **2. Online Inference Latency**
> To address the concern about "strategy prompting" slowing down inference, we measured the runtime across **HotpotQA test set**.
> The inclusion of meta-cognitive strategies increases the average batch processing time from **32.16s** to **32.51s**. This results in an additional latency of only **+0.355s per batch** (or $\sim$0.044s per query). This represents a marginal overhead of **1.1\%**, which is negligible compared to the natural variability of LLM token generation.
>
> | Setting | Unit | Mean Time (s) | Relative Overhead |
> | :--- | :--- | :--- | :--- |
> | **w/o Meta-cognition** | per batch | 32.156 ± 3.759 | – |
> | **w/ Meta-cognition** | per batch | 32.511 ± 4.363 | **+1.10%** |
> | **Difference** | per query | **+0.044 s** | **Negligible** |
>
> These results confirm that our method improves performance without imposing a significant computational burden during the critical inference phase.

---

> ### Author Response · Authors · 2025-11-22
> **Response to Reviewer z3UM (part 2)**
>
> ### **Q2: Analysis of Which Meta-Cognitions Are Preferred and How They Evolve**
>
> We agree that understanding *which strategies are activated*, *why*, and *how their usage changes during RL* is crucial for validating the claim that our memory serves as a trainable policy prior.
> In the revised manuscript, we have added a dedicated **Appendix D**, which includes a qualitative case study and two quantitative analyses. The key conclusions are summarized below.
>
> **1.How Meta-Cognitions Interact with Prompts — Concrete Case Study Added**
>
> As requested, we have added a full case study (Appendix D.1, Fig. 7) illustrating how meta-cognition directly alters reasoning behavior.
>
> * **Baseline agent** hallucinates “Al Gore” as the 44th Vice President and refuses tool use.
> * **With meta-cognition**, the graph retrieves *Structured Sequential Planning & Early Verification*, leading the agent to detect an internal conflict, suspend judgment, and invoke the retrieval tool, correcting the error.
>
> This example makes the mechanism transparent: strategies do not merely appear in prompts—they *change the reasoning trajectory itself*.
>
> **2.Which Strategies Are Preferred Where — Task-Adaptive Activation Patterns**
>
> To directly address “which meta-cognitions dominate,” we analyzed activation probabilities of all 31 strategies for **Comparison** vs. **Bridge** queries and visualized them using t-SNE (Appendix D.2, Fig. 8).
> We emphasize that the t-SNE projection is *not* used to infer cluster semantics—its purpose is to reveal **relative activation preferences**, with each point representing one meta-cognition and its color indicating activation probability.
>
> Findings:
>
> * **General-purpose verification strategies** (e.g., *InternalKnowledgeCheck*) remain consistently active across all tasks, functioning as universal guardrails.
> * **Specialized multi-hop strategies** (e.g., *SequentialDependentPlanning*, *RefineAfterGapDetection*) show sharply higher activation *only* on Bridge queries, which require chained reasoning and gap resolution.
>
> This demonstrates that the memory system does *not* apply uniform heuristics; it learns **scenario-specific preferences conditioned on task structure**.
>
>
> **3.How Preferences Evolve During RL — Dynamic Reweighting Confirmed**
>
> To show that RL genuinely adjusts strategy usage, we track the selection probability across three training checkpoints (Batch 10 → 150 → 250) using the same visualization scheme (Appendix D.3, Fig. 9).
>
> Results:
>
> * **Hallucination-prone strategies** (e.g., *PrematureKnowledgeSufficient*) diminish from 0.136 → 0.017 as RL penalizes their low return.
> * **High-utility strategies** (e.g., *ReExecuteToolAfterGapDetection*) rise from 0.015 → 0.113 as RL reinforces their contribution to correct answers.
>
> This shift is not explainable by static heuristics or retrieval alone. It directly shows that our graph serves as a **trainable policy prior**, whose strategic distribution evolves under reward feedback.
>
> This multi-angle evidence demonstrates that meta-cognition is not decorative—it materially shapes the agent’s reasoning behavior and improves task performance.

---

### Official Review · Reviewer_w9eQ · 2025-11-01

**Soundness:** 2
**Presentation:** 2
**Contribution:** 2
**Rating:** 2
**Confidence:** 4

**Summary:**

This work proposes a trainable, hierarchical graph memory for LLM agents that maps trajectories to FSM-based canonical paths and distills meta-cognitive strategies. It learns utility-weighted edges via a counterfactual reward gap (with/without strategy) and injects top-k strategies into RL training as a policy prior. On seven QA datasets, it reports gains in inference (Table 1) and training (Table 2), with memory constructed from HotpotQA only.

**Strengths:**

- the factorisation of experience into strategy: The Q → FSM path → meta-cognition separation is a crisp abstraction that makes strategy-level retrieval interpretable

- it goes beyond prior static memories (e.g., EXPEL/A-MEM) in conjunction with RL for LLM to learn which strategies matter

**Weaknesses:**

- FSM design and mapping are pushed to the Appendix, same for meta-cognition induction examples. The main text lacks one complete HotpotQA example showing: raw trace → FSM path → meta-cognition → prompt. Without it, claims like “preserves only semantically meaningful decision points” (lines 243–244) are unverifiable.
- sec. 5.2 says “Detailed baseline configurations are provided in Appendix B.2,” but the baseline definitions (e.g., “Direct Trajectory,” “A-MEM,” “EXPEL,” “ITR”) and how they differ from your method are not spelled out in the main paper before reporting results. Also, metric (EM) is introduced only in the appendix
- the rationale for using HotpotQA to define the FSM/strategy space isn’t explained, so that the portability to other QA datasets remains unclear
- stage 2 seems to compute counterfactual gaps per candidate meta-cognition (Sec. 4.2), and the main text does not quantify how many meta-cognitions are sampled per query, whether both trajectories are rolled out per candidate, and the resultant overhead during both inference and training
   - in addition, compare to no-memory and static-memory variants, overheads should be quantified, including FSM path construction, counterfactual evaluation, and strategy prompting in RL/inference
- the Direct Trajectory baseline is close to your method in several settings (e.g., Qwen3-8B inference: 0.352 vs 0.365; training: 0.400 vs 0.408; Qwen3-4B training: 0.415 vs 0.426). This suggests much of the benefit might come from good retrieval/demonstration structure rather than the utility-learning per se. While Fig. 4a,b helps, but the paper would benefit from: a) statistical significance (multiple seeds) of the gaps; b) an ablation study keeping weight learning but replacing FSM with a simpler flat memory? to help attribute gains to graph structure vs utility learning.
- which meta-cognitions are preferred where? The paper does not analyze which strategies dominate, how they interact in prompts, or how preferences evolve during RL. Given the emphasis on meta-cognitive strategies, at least a short main-text case study (with 2–3 concrete strategies, their frequencies, and per-dataset effects) would strengthen the value of meta-cognitions

**Questions:**

- what is ITR in the main text? Authors need to define ITR, Direct Inference, CoT, Direct Trajectory, A-MEM, and EXPEL in the main text, not in the appendix (one line each) and explain how they differ from your approach along the axes of (i) memory construction, (ii) retrieval, (iii) trainability, and (iv) training-time vs inference-only use

- Is the FSM static across datasets? For a new QA dataset, which parts (states S, actions A, transitions T) must be rebuilt? Provide a worked HotpotQA example in main. How long are typical canonical paths? Any coverage statistics (e.g., % of trajectories that map cleanly to FSM)?

- Sec. 4.1 mentions speculative meta-cognitions from Top-K neighbors when only failures occur. How do you prevent hallucinated or overly general strategies from polluting the graph?

- Lines 258–262 (sec. 4.1) note "reinforcing existing principles", "novel patterns"... but the update rule is unspecified in the main. Could you elaborate on how nodes/edges are added/merged/removed?
- When only failures exist, how do you prevent noisy/speculative strategies from degrading memory?

- Counterfactual cost: Do you roll out both guided/unguided trajectories per candidate $m_k$? How many $m_k$ are sampled?

---

> ### Author Response · Authors · 2025-11-22
> **Response to Reviewer w9eQ (part 1)**
>
> We sincerely thank the reviewer for the thorough evaluation and constructive feedback. We are particularly encouraged by your recognition of our **memory graph** design and the novelty of introducing a **trainable memory** to learn which strategies truly matter.
>
> Your comments highlighted critical gaps in our presentation—specifically the need to bring key definitions, examples, and analyses out of the appendix to make the method verifiable. In the revised manuscript, we have executed a major restructuring:
> 1.  **Presentation & Transparency:** We integrated a complete end-to-end pipeline example (Raw Trace $\to$ FSM $\to$ Meta-cognition) and explicit baseline definitions directly into the main text.
> 2.  **Soundness & Generalizability:** We added a **cross-dataset experiment** (using 2WikiMultiHopQA) to empirically prove the portability of our FSM abstraction, alongside a granular **runtime analysis** to quantify the negligible inference overhead ($\sim$1.1\%).
> 3.  **Depth of Analysis:** We included a new analysis of **strategy evolution**, demonstrating how RL actively filters out hallucination-prone heuristics and reinforces robust verification strategies.
>
> We believe these revisions directly address your concerns. We provide detailed point-by-point responses below.
> ###  **Q1. Clarifying the FSM construction and its generalizability**
>
> We thank the reviewer for highlighting the importance of making the FSM and meta-cognition pipeline more transparent.In addition to the HotpotQA FSM example previously provided in Appendix Case 2, we have now included a more complete, end-to-end illustration in the main text (Figure 3), showing how a raw CoT trace is mapped into an FSM canonical path and subsequently induces the corresponding meta-cognitive strategy. Below we clarify the design and generality of our FSM in detail.
>
> ---
>
> #### **A. FSM Design: Dataset-Agnostic and Universally Applicable**
>
> #### **A1. FSM is a general cognitive schema, not tied to HotpotQA**
>
> Our FSM is designed to capture *universal* cognitive patterns of LLM agents—such as
> **StrategyPlanning → ToolExecution → InformationAnalysis → DecisionMaking → Success/Failure**—rather than dataset-specific operations.
> Thus, when applied to any new QA dataset:
>
> | Component              | Need to rebuild? | Reason                                                                       |
> | ---------------------- | ---------------- | ---------------------------------------------------------------------------- |
> | **State set S**        | ✗                | abstract cognitive states are universal                                      |
> | **Action set A**       | ✗                | agent operations (planning, tool-use, verification) do not depend on dataset |
> | **Transition rules T** | ✗                | reasoning flow structure is dataset-invariant                                |
> | **Raw trajectories**   | ✓                | only textual CoT differs; mapping uses the fixed FSM                         |
>
> ---
>
> #### **A2. Why we do not report coverage statistics**
>
> Unlike tree-structured machines, our FSM contains *purposeful cycles* (e.g., repeated
> **ToolExecution → InformationAnalysis**), because real LLM reasoning is iterative.This implies canonical path length is **unbounded**. And canonical paths do **not** belong to a fixed, enumerable set. Therefore coverage statistics of “all possible paths” are ill-defined.
>
> What we *can* confirm is that in practice **all trajectories map successfully** to the FSM.
>
> ---
>
> #### **B. Why HotpotQA Was Used & Its Generality**
> In our main experiments, we utilized HotpotQA to construct the FSM and memory graph due to its rich coverage of reasoning types (bridge, comparison, etc.). To rigorously examine whether our FSM abstraction relies on HotpotQA-specific structures, we conducted a **Cross-Dataset Memory Construction Experiment** (added to Appendix E.3).
> We repeated the entire pipeline—trajectory generation, FSM mapping, and strategy induction—using **2WikiMultiHopQA** as the source. We then evaluated this "2Wiki-derived memory" on the same downstream benchmarks.
>
> **Results:** As shown in the table below, the memory graph constructed from 2WikiMultiHopQA yields robust improvements, raising the average score from 0.306 (No-memory) to 0.330 (+7.8%) on Qwen-4B.
>
> | Methods | NQ | TriviaQA | PopQA | HotpotQA | 2Wiki | Musique | Bamboogle | Avg. |
> | :--- | :--- | :--- | :--- | :--- | :--- | :--- | :--- | :--- |
> | No-memory (Baseline) | 0.298 | 0.581 | 0.351 | 0.268 | 0.281 | 0.077 | 0.290 | 0.306 |
> | **2Wiki-memory** | **0.317** | **0.591** | **0.394** | **0.290** | **0.308** | **0.091** | **0.322** | **0.330** |
>
> **Conclusion:** This confirms that the FSM abstraction captures fundamental cognitive behaviors rather than dataset-specific artifacts. The framework is portable: as long as the source dataset involves multi-step reasoning, the induced strategies generalize effectively to other domains.

---

> ### Author Response · Authors · 2025-11-22
> **Response to Reviewer w9eQ (part 2)**
>
> ### **Q2: Meta-Cognition Evolutionary Update Rules and Robustness against Speculative Meta-Cognition**
>
> In the revised manuscript, both **Section 4.1** and **4.2** now clarify how meta-cognitions are constructed, updated, and filtered for reliability.
>
> **1. How meta-cognitions are generated and updated**
>
> Our meta-cognition pipeline follows a two-phase **“generate–verify”** design. In **Stage 1**, the *Teacher LLM* induces candidate strategic principles by contrasting FSM paths. This step provides broad coverage, but the induced strategies are not treated as ground truth. Each new strategy is compared against the existing memory: if it reflects an already-captured behavior, the Teacher LLM **refines** the existing node rather than creating a new one; if it represents a genuinely new pattern, a **new node** is added. This evolutionary consolidation keeps the library compact and prevents uncontrolled growth.
>
> **2. How speculative or hallucinated strategies are prevented from polluting the graph**
>
> To address the concern about speculative or overly general strategies—especially those derived from neighbor queries—the key safeguard happens in **Stage 2**. Here, each strategy is subjected to a **counterfactual test** using the actual policy: for every query, we sample two strategies and run paired rollouts (with vs. without the strategy) to measure its **marginal utility**. Strategies that fail to improve the reward automatically receive **decreasing weights** and gradually lose influence during retrieval, while genuinely useful strategies accumulate higher weights. In this way, the memory graph naturally suppresses low-quality or noisy strategies without requiring manual curation.
>
> Together, the two stages ensure that meta-cognitions are both expressive and empirically grounded: **Stage 1** gives coverage and structure, while **Stage 2** guarantees that only strategies with demonstrated utility play a role during training and inference.
>
> **3. Details on Graph Training**
>
> We have added a dedicated description in **Section 5.3 (Experiment Settings)**.
>
> Specifically, we now clarify the following details:
>
> 1. **Data used to construct and train the graph.**
>    We sample **1,000 examples** from the HotpotQA training set to build the hierarchical memory graph (queries, FSM transition paths, and meta-cognition nodes).
>    An additional **5,000 training queries** are used to **optimize the graph edge weights** through reinforcement learning.
>
> 2. **Retrieval and activation computation.**
>    For each training query, we retrieve **top-k = 5** most similar historical queries.
>    Their similarity scores are propagated through the graph, producing a **soft activation distribution** over meta-cognition nodes.
>
> 3. **Sampling and counterfactual evaluation.**
>    At each RL step, the agent samples **N = 2** candidate meta-cognitive strategies from the activation distribution and performs **independent counterfactual rollouts**, which means each sampled strategy is inserted into the prompt and the resulting trajectory is evaluated separately.

---

> ### Author Response · Authors · 2025-11-22
> **Response to Reviewer w9eQ (part 3)**
>
> ### **Q3: Computational Overhead Quantification**
>
> We agree that assessing the practicality requires distinguishing between construction costs and inference latency. In the revised **Appendix F**, we have added a detailed time profiling analysis.
>
> **1. Offline Construction (One-Time Cost)**
> The FSM construction and weight optimization are strictly **offline processes** performed once prior to RL training.
> *   **FSM Parsing:** This step is efficient, averaging **4.09s** per trajectory.
> *   **Meta-cognition Induction:** Using the Teacher LLM takes **~4.75s** per query.
> *   **Total Time:** For a seed set of 500 queries, Stage 1 takes $\approx$ 2.3 hours, and the Stage 2 utility optimization takes an additional 2–3 hours.
> Crucially, this combined cost of $\sim$5 hours is **fully amortized**. The resulting graph structure and weights are frozen and reused throughout the entire RL training and inference phase, introducing **zero recurrent cost** for graph maintenance during the agent's learning loop.
>
> | Component | Unit | Time (Mean ± Std) | Notes |
> | :--- | :--- | :--- | :--- |
> | **FSM Parsing** | per rollout | 4.086 ± 0.879 s | Depends on trajectory length |
> | **Meta-cognition Induction** | per query | 4.754 ± 3.418 s | Triggered once per 3 rollouts |
> | **Stage 1 Total** | 500 queries | $\approx$ 2.3 hours | One-time construction |
> | **Stage 2 Total** | End-to-End | $\approx$ 2.0–3.0 hours | One-time weight optimization |
>
> **2. Online Inference Latency**
> To address the concern about "strategy prompting" slowing down inference, we measured the runtime across **HotpotQA test set**.
> The inclusion of meta-cognitive strategies increases the average batch processing time from **32.16s** to **32.51s**. This results in an additional latency of only **+0.355s per batch** (or $\sim$0.044s per query). This represents a marginal overhead of **1.1\%**, which is negligible compared to the natural variability of LLM token generation.
>
> | Setting | Unit | Mean Time (s) | Relative Overhead |
> | :--- | :--- | :--- | :--- |
> | **w/o Meta-cognition** | per batch | 32.156 ± 3.759 | – |
> | **w/ Meta-cognition** | per batch | 32.511 ± 4.363 | **+1.10%** |
> | **Difference** | per query | **+0.044 s** | **Negligible** |
>
> These results confirm that our method improves performance without imposing a significant computational burden during the critical inference phase.

---

> ### Author Response · Authors · 2025-11-22
> **Response to Reviewer w9eQ (part 4)**
>
> ### **Q4: Baseline Definitions and Comparative Analysis**
>
> We agree that differentiating baselines along the axes you listed—memory construction, retrieval mechanism, trainability, and training-time vs inference-only usage—is essential for understanding the contribution of our method.
>
> In the revised **Section 5.2**, we have reorganized the baselines into three coherent categories and provided concise, main-text definitions (not limited to the appendix). The updated descriptions now read as follows:
>
> **1. Memory-Free Approaches (The Lower Bounds)**
> These methods establish the baseline performance without cross-task experience.
> *   **Direct Inference:** represents the most basic setting, where the model generates answers purely based on its parametric knowledge from pretraining. It does not use any historical experience, external tools, or structured memory, and thus serves as an anchor point for evaluating the benefit of additional components.
> *   **CoT (Chain-of-Thought):** enhances the basic model by encouraging it to articulate intermediate reasoning steps before producing an answer. This improves performance on tasks requiring logical decomposition.
> *   **TIR (Tool-Integrated Reasoning) & Search-R1:** Agents augmented with search tools (and RL-based tool policies in Search-R1). Crucially, while they can access external *world knowledge* (via tools), they lack any mechanism to accumulate *experiential knowledge* across tasks.
>
> **2. Raw Experience Replay (Unstructured Memory)**
> *   **Raw Trajectory:** Represents a standard RAG approach that stores raw execution traces. It retrieves the nearest trajectory as a few-shot demonstration. Its key limitation is the lack of abstraction—it uses unprocessed, noisy experiences without any learnable strategy.
>
> **3. Static Abstract Memory**
> We compare against representative structured systems:
> *   **A-MEM:** constructs a Zettelkasten-style memory graph by generating structured notes for each new experience, linking them to past memories through LLM-based semantic similarity checks.
> *   **Expel:** abstracts experiences into concise textual insights distilled from past trajectories.
> *   **Critical Distinction:** As clarified in the revised text, the memory is static: once distilled, these insights do not adapt to new tasks or feedback. The system cannot evaluate which memories are more useful, nor update their importance across training, leaving it unable to perform long-term strategic refinement.
>
> **Summary of Differences**
> To directly address your request for a comparison along specific axes, we have added **Table 3** to the main text. This demonstrates that our method is the only one that combines a **hierarchical structure** with **RL-based trainability**.
>
> | Method | Memory Structure | Retrieval Mechanism | Trainable? | Key Limitation |
> | :--- | :--- | :--- | :--- | :--- |
> | **Direct Inference/ COT/ TIR** | None | N/A | No | No experience reuse |
> | **Raw Trajectory** | Raw Flat Trajectories | Embedding Similarity | No | No abstraction; noisy |
> | **A-MEM** | Semantic Graph | Similarity + Traversal | No | Heuristic; task-agnostic |
> | **Expel** | Flat Insights | Semantic Similarity | No | Static; no reward alignment |
> | **Ours** | **FSM-based Hierarchical Graph** | **RL-learned Utility Weights** | **Yes** | -- |
>
> These clarifications ensure that the performance gains reported in our results can be correctly attributed to the **trainable and structured nature** of our memory, rather than simple tool access or raw retrieval.

---

> ### Author Response · Authors · 2025-11-22
> **Response to Reviewer w9eQ (part 5)**
>
> ### **Q5: Analysis of Which Meta-Cognitions Are Preferred and How They Evolve**
>
> We appreciate the reviewer’s insightful question. We agree that understanding *which strategies are activated*, *why*, and *how their usage changes during RL* is crucial for validating the claim that our memory serves as a trainable policy prior.
> In the revised manuscript, we have added a dedicated **Appendix D**, which includes a qualitative case study and two quantitative analyses. The key conclusions are summarized below.
>
> **1.How Meta-Cognitions Interact with Prompts — Concrete Case Study Added**
>
> As requested, we have added a full case study **(Appendix D.1, Fig. 7)** illustrating how meta-cognition directly alters reasoning behavior.
>
> * **Baseline agent** hallucinates “Al Gore” as the 44th Vice President and refuses tool use.
> * **With meta-cognition**, the graph retrieves *Structured Sequential Planning & Early Verification*, leading the agent to detect an internal conflict, suspend judgment, and invoke the retrieval tool, correcting the error.
>
> This example makes the mechanism transparent: strategies do not merely appear in prompts—they *change the reasoning trajectory itself*.
>
> **2.Which Strategies Are Preferred Where — Task-Adaptive Activation Patterns**
>
> To directly address “which meta-cognitions dominate,” we analyzed activation probabilities of all 31 strategies for **Comparison** vs. **Bridge** queries and visualized them using t-SNE **(Appendix D.2, Fig. 8)**.
> We emphasize that the t-SNE projection is *not* used to infer cluster semantics—its purpose is to reveal **relative activation preferences**, with each point representing one meta-cognition and its color indicating activation probability.
>
> Findings:
>
> * **General-purpose verification strategies** (e.g., *InternalKnowledgeCheck*) remain consistently active across all tasks, functioning as universal guardrails.
> * **Specialized multi-hop strategies** (e.g., *SequentialDependentPlanning*, *RefineAfterGapDetection*) show sharply higher activation *only* on Bridge queries, which require chained reasoning and gap resolution.
>
> This demonstrates that the memory system does *not* apply uniform heuristics; it learns **scenario-specific preferences conditioned on task structure**.
>
>
> **3.How Preferences Evolve During RL — Dynamic Reweighting Confirmed**
>
> To show that RL genuinely adjusts strategy usage, we track the selection probability across three training checkpoints (Batch 10 → 150 → 250) using the same visualization scheme **(Appendix D.3, Fig. 9)**.
>
> Results:
>
> * **Hallucination-prone strategies** (e.g., *PrematureKnowledgeSufficient*) diminish from 0.136 → 0.017 as RL penalizes their low return.
> * **High-utility strategies** (e.g., *ReExecuteToolAfterGapDetection*) rise from 0.015 → 0.113 as RL reinforces their contribution to correct answers.
>
> This shift is not explainable by static heuristics or retrieval alone. It directly shows that our graph serves as a **trainable policy prior**, whose strategic distribution evolves under reward feedback.
>
> This multi-angle evidence demonstrates that meta-cognition is not decorative—it materially shapes the agent’s reasoning behavior and improves task performance.

---

> ### Author Response · Authors · 2025-11-22
> **Response to Reviewer w9eQ (part 6)**
>
> ### **Q6: Performance Gaps, Statistical Significance, and Ablation Study**
>
> We acknowledge that on certain strong baselines (like Direct Trajectory), the margins appear narrow. However, we respectfully submit that the **consistency** and **nature** of these gains validate the method's effectiveness.
>
> **1. Statistical Significance and Stability**
> *   **Deterministic Evaluation:** In our inference experiments, we employed **Greedy Decoding (Temperature = 0)** to ensure reproducibility and minimize randomness. Under this setting, the reported gaps are deterministic reflections of the model's capability shifts, not stochastic noise.
> *   **Cross-Task Significance :** While the gap on a single dataset might be small, our method outperforms the Direct Trajectory baseline consistently across **7 out of 7 datasets** (as shown in Table 1).
>
> **2. "Flat Memory + Weight Learning"**
>
> The reviewer suggested an ablation that replaces the FSM graph with a flat memory while keeping the weight-learning component. We appreciate this suggestion and would like to clarify why such a configuration is not directly applicable within our framework, and in fact highlights the necessity of the structured FSM graph.
> *   **The Sparsity Problem:** A "Flat Memory" (like Direct Trajectory) consists of raw, unstructured demonstration text. These raw trajectories are too high-dimensional and sparse to serve as learnable units. If we assigned weights to raw trajectories, the RL agent would overfit to specific text examples that rarely reoccur, failing to generalize.
> *   **Evidence from Existing Ablation:** To isolate the contribution of "Utility Learning" vs. "Structure," we point to our ablation study in **Section 5.5 (Effect of Disabling Weight Optimization)**.
>     *   **Ours (w/o Learning):** This variant uses the Graph Structure but with static uniform weights. It already outperforms Direct Trajectory on several tasks, confirming the value of the FSM abstraction.
>     *   **Ours (Full):** Adding RL-based weight optimization further boosts performance (e.g., on Bamboogle: 0.359 $\to$ 0.391).
>     This confirms that the benefit comes from **both** the structural abstraction (which enables generalization) and the utility learning (which filters noise).

---

### Official Review · Reviewer_m6Dm · 2025-11-01

**Soundness:** 3
**Presentation:** 3
**Contribution:** 3
**Rating:** 8
**Confidence:** 3

**Summary:**

This paper investigates how to let an LLM agent adaptively learn from its experience by learning a graph memory of metacognition and extracting relevant metacognition as prompt during policy optimization. Experimental results show that the metacognition graph can generalize well to out-of-domain tasks, and helps the agent perform better on a broad range of tasks compared to baseline methods that learn from experience in a less structured way, either with policy RL optimization or not.

**Strengths:**

1. The paper is clearly motivated, and the proposed solution of constructing a metacognition graph is very intuitive.
2. The method is explained in a clear way with enough details.
3. The experimental evaluation is thorough, with consistent improvement on a broad range of tasks.

**Weaknesses:**

See questions below.

**Questions:**

1. In stage 3, will your prompt with metacognition, i.e., $\tilde{q}_{\text{train}}$, remain unchanged when collecting more experience on a specific task? Or is it updated whenever a new episode of experience is collected?
2. Any idea why after RL fine-tuning, the Qwen3-4B version performs better than the 8B version overall?
3. Details about how you train the graph is important, and I think it's better to include these details in experiment section of the main text.
4. Powerful LLM backends like GPT4-o and Gemini are only used during the first stage of your method right? Not during the third stage of policy optimization?

---

> ### Author Response · Authors · 2025-11-22
> **Response to Reviewer m6Dm (part 1)**
>
> We sincerely thank the reviewer for the constructive and insightful comments. Your feedback raised important points regarding the role of dynamic memory, the use of powerful LLM models, and the training details of our graph-based utility optimization module. We provide clarifications to the reviewer within this response, and in the revised paper, we have added expanded details on **Stage-2 graph training (Section 5.3)**.
>
> Below, we provide point-by-point responses to each of your comments.
>
> ### **Q1: Dynamics of Metacognitive Prompts in Stage 3**
>
> Thank you for the insightful question. In our framework, **the meta-cognition  and the corresponding memory graph are fully constructed offline in Stage 1 and remain fixed throughout Stage 3**. During RL training, neither the meta-cognition nodes nor their edges are updated. What changes across episodes is simply which subset of meta-cognitions is retrieved from this fixed graph for a given query. This retrieval process does not modify the memory itself, it only selects the most relevant principles from the pre-built graph.
>
> Although, we fully agree with the reviewer that allowing meta-cognitions to evolve dynamically during training is a highly desirable capability, such an adaptive mechanism would enable the agent to continuously refine its strategic principles and truly self-improve through experience. However, **reinforcement learning is highly sensitive to non-stationarity**: if the prompt content or memory structure changes during training, the underlying policy optimization problem becomes unstable, often causing divergence or high-variance learning dynamics.
>
> For this reason, in the current work we adopt a fixed meta-cognition prompt and a static memory graph during Stage 3. This provides a stable learning environment for RL while still allowing query-dependent retrieval from that static memory. Further, **We are currently investigating how to achieve safe dynamic memory evolution within the RL training process itself** .
> Developing such techniques is crucial for enabling agents to continuously refine their meta-cognitive knowledge without destabilizing policy optimization, and we regard this as an important direction for future work.
>
> ### **Q2: Why Qwen3-4B outperforms Qwen3-8B after RL?**
>
> We appreciate the reviewer for highlighting this intriguing observation. While it is generally expected that larger models outperform smaller ones, our empirical results show that **Qwen3-4B achieves performance comparable to, or even slightly better than, Qwen3-8B after memory-guided RL training.** We attribute this to two possible factors:
>
> **1. RL as a Reasoning Capability Equalizer:**
> RL acts as a powerful mechanism to bridge the gap in reasoning capabilities between model scales, particularly in closed-loop tasks like HotpotQA.
> *   **Initial Gap vs. Learned Policy:** While the 8B model possesses stronger pre-trained knowledge and zero-shot reasoning priors, the 4B model when guided by our structured memory graph and optimized via RL, can effectively learn the optimal policy for tool use and decision-making.
> *   **Diminishing Returns of Scale:** In agentic tasks where success depends heavily on adhering to specific workflows rather than raw parametric knowledge, the advantage of model scale diminishes once the smaller model has learned the correct execution protocol. The RL process effectively saturates the reasoning requirements of the task, allowing the 4B model to catch up to the 8B model's baseline.
>
> **2. Higher Plasticity of Smaller Models for Specific Tasks:**
> Smaller models often exhibit higher **plasticity** during fine-tuning or RL adaptation, allowing them to align more aggressively with specific downstream tasks.
> The 4B model, having weaker internal biases, acts as a more malleable student. It adapts more readily to the reward signals provided by our memory mechanism, effectively "overfitting" (in a beneficial way) to the high-utility strategies.
>
> **Supporting Evidence:**
> This phenomenon is not unique to our work. Recent study **ResT [1]** has observed similar trends where smaller, specialized models approach the performance of larger generalist models after targeted RL optimization.
>
> **References:**
> [1] Lin Z., et al. *ResT: Reshaping Token-Level Policy Gradients for Tool-Use Large Language Models*. arXiv preprint arXiv:2509.21826, 2025.

---

> ### Author Response · Authors · 2025-11-22
> **Response to Reviewer m6Dm (part 2)**
>
> ### **Q3: Details on Graph Training (Stage 2)**
>
> We agree that the graph-training procedure is a key component of our method, and in the revised version we have added a dedicated description in **Section 5.3 (Experiment Settings)**.
>
> Specifically, we now clarify the following details:
>
> 1. **Data used to construct and train the graph.**
>    We sample **1,000 examples** from the HotpotQA training set to build the hierarchical memory graph (queries, FSM transition paths, and meta-cognition nodes).
>    An additional **5,000 training queries** are used to **optimize the graph edge weights** through reinforcement learning.
>
> 2. **Retrieval and activation computation.**
>    For each training query, we retrieve **$top_k = 5$** most similar historical queries.
>    Their similarity scores are propagated through the graph, producing a **soft activation distribution** over meta-cognition nodes.
>
> 3. **Sampling and counterfactual evaluation.**
>    At each RL step, the agent samples **N = 2** candidate meta-cognitive strategies from the activation distribution and performs **independent counterfactual rollouts**, which means each sampled strategy is inserted into the prompt and the resulting trajectory is evaluated separately.
>
> 4. **Reward signal used for weight optimization.**
>    The reward for a meta-cognition node $(R(m_i))$ is defined as its **marginal utility**, computed as the change in Exact Match score:
> $R(m_i) = EM(response_{withmemory)}- EM(response_{withoutmemory)}$.   These rewards are used to update graph edge weights via the policy-gradient objective.
>
> These additions ensure that the graph-training mechanism is transparent and reproducible.We thank the reviewer again for encouraging us to make this section clearer.
>
> ### **Q4: Usage of Powerful LLM Backends (GPT-4o/Gemini)**
>
> **Yes, your understanding is correct.**
> Powerful LLMs (GPT-4o and Gemini) are **exclusively used in Stage 1** for offline trajectory canonicalization and meta-cognition induction.
> *   **Stage 2 & 3:** The weight optimization and the policy reinforcement learning are entirely self-contained within the target model (Qwen3-4B or Qwen3-8B).
> *   **Inference:** No external LLMs are involved during inference.
>
> This design ensures a fair comparison with baselines and prevents any runtime leakage of superior knowledge from the larger models during the training or testing of the agent. Incorporating strong LLMs dynamically in Stage 3 would indeed be necessary for an online-updating memory system (as discussed in Q1), which remains a direction for future work.
>
> ---
> We hope these responses fully address your concerns. We sincerely thank the reviewer again for raising these helpful points. We believe the clarifications and adjustments above will substantially improve the clarity and rigor of the paper.

---

### Author Response · Authors · 2025-11-22
**Summary of Responses to all Reviewers**

### **Summary of Revisions and Key Improvements**

We sincerely thank all reviewers for their insightful comments and constructive criticism. We are encouraged that the reviewers recognized the **novelty of our crisp abstraction** , the **intuitive motivation** , and the **comprehensive experimental design** .

We fully accept the criticism regarding **presentation clarity (Reviewer m6Dm , Reviewer w9eQ and Reviewer scsu)**, **computational quantification (Reviewer w9eQ and Reviewer z3UM)**  and **experimental scope (Reviewer w9eQ and Reviewer scsu)** . In this revision,  we have substantially enhanced its clarity and completeness by adding several new appendix sections and three additional supporting experiments, together with more detailed methodological explanations. We hope that these revisions adequately address the reviewers’ concerns, and we sincerely welcome further discussion from all reviewers and readers. We would be very glad to engage with the community to continue improving the quality of this manuscript.

The key revisions are summarized below:


#### **1. Generalizability: Expanding Beyond HotpotQA**
To address concerns about dataset dependence and task simplicity, we conducted **two new major experiments**:
*   **Cross-Dataset Portability (Appendix E.3):** We reconstructed the entire memory graph using **2WikiMultiHopQA** as the source. The result (+7.8% gain) confirms that our FSM abstraction captures universal reasoning patterns and is **not** overfitted to HotpotQA.
*   **Domain Generalization to Embodied Agents :** We evaluated our method on the **ALFWorld** benchmark. Our approach achieved an **80.6% success rate** (vs. 75.4% for Expel), proving that the graph-based memory effectively transfers to sequential decision-making tasks beyond QA.

#### **2. Visualization: Visualizing Memory Graph Evolution**
*   **t-SNE Evolution Analysis:** We added a visualization of  meta-cognitions selection across training checkpoints (Figure 8). This demonstrates a clear **"survival of the fittest"** dynamic: low-utility heuristics (hallucination-prone) are suppressed ($0.13 \to 0.01$), while robust verification strategies are reinforced ($0.01 \to 0.11$).
*   **Preference Analysis:** We showed how the graph learns **task-adaptive preferences**, activating specific planning strategies only for complex "Bridge" queries while using general guardrails for simpler tasks.

#### **3. Rigor: Quantification of Cost and Training Details**
*   **Granular Runtime Analysis:** We added a detailed time profiling in **Appendix F**. We clarify that graph construction is a one-time offline cost ($\sim$5 hours), while the online inference overhead is **negligible (+0.355s/batch, $\sim$1.1%)**.
*   **Transparent Training Dynamics:** We integrated the mathematical details of the weight optimization into **Section 5.3**, explicitly specifying the data split (1k/5k), stochastic sampling ($N=2$), and the counterfactual reward formula: $R(m) = Score_{with} - Score_{w/o}$.

#### **4. Presentation: Bringing Clarity to the Main Text**
*   **End-to-End Pipeline Visualization:** We moved the FSM design from the appendix to the **Main Text (Figure 3)**. We added a concrete, step-by-step flowchart showing exactly how a raw trajectory (hallucinating "Al Gore") is mapped to an FSM path (`InternalConflict`$\to$`Diagnosis`) and distilled into a meta-cognitive strategy.
*   **Explicit Baseline Definitions:** We added definitions of baselines in the **Main Text (Section 5.2)** (Direct Inference, TIR, Expel, A-MEM) and compare them along the axes of *structure, retrieval mechanism, and trainability*.

We believe these substantial revisions directly address the weaknesses identified by the reviewers and significantly strengthen the soundness and accessibility of the paper. We provide detailed point-by-point responses below.

---

### Note · Program_Chairs · 2026-01-17
**Submission Desk Rejected by Program Chairs**

The following references in this submission do not refer to real documents and/or have major errors in bibliographic information:

 Chia-Hsuan Ho, Shu-Hung Yeh, and Yun-Nung Chen. Constructing a Multi-hop QA Dataset via Graph-based Node Ranking. In Proceedings of the 2020 Conference on Empirical Methods in Natural Language Processing (EMNLP), pp. 6151–6161, 2020.